# Biological Scaffolds Assembled with Magnetic Nanoparticles for Bone Tissue Engineering: A Review

**DOI:** 10.3390/ma16041429

**Published:** 2023-02-08

**Authors:** Zheng Li, Le Xue, Peng Wang, Xueqian Ren, Yunyang Zhang, Chuan Wang, Jianfei Sun

**Affiliations:** 1State Key Laboratory of Bioelectronics, Jiangsu Key Laboratory of Biomaterials and Devices, School of Bioscience and Medical Engineering, Southeast University, Nanjing 210009, China; 2Clinical Medical Engineering Department, The Affiliated Zhongda Hospital of Southeast University Medical School, Nanjing 210009, China; 3Center of Modern Analysis, Nanjing University, Nanjing 210000, China; 4Naval Medical Center of PLA, Naval Medical University (Second Military Medical University), Shanghai 200433, China

**Keywords:** superparamagnetic iron oxide nanoparticles, biological assembly, biological scaffolds, bone repair

## Abstract

Superparamagnetic iron oxide nanoparticles (SPION) are widely used in bone tissue engineering because of their unique physical and chemical properties and their excellent biocompatibility. Under the action of a magnetic field, SPIONs loaded in a biological scaffold can effectively promote osteoblast proliferation, differentiation, angiogenesis, and so on. SPIONs have very broad application prospects in bone repair, bone reconstruction, bone regeneration, and other fields. In this paper, several methods for forming biological scaffolds via the biological assembly of SPIONs are reviewed, and the specific applications of these biological scaffolds in bone tissue engineering are discussed.

## 1. Introduction

Bone defect repair has always been an important research topic in the field of trauma medicine and biomaterials. Clinically, the repair of large-scale bone defects has not been effectively solved [1,2]. So far, it has not been possible to produce an ideal artificial material as a substitute for bone transplantation. In clinical practice, the three most commonly used methods are autogenous bone transplantation, allogeneic bone transplantation, and artificial biological substitute materials [3,4]. Among them, the most common application is still autogenous bone transplantation, which is regarded as the “gold standard” of bone transplantation [5,6]. However, this method also has some limitations, such as an insufficient supply of transplant materials, complications caused by donors, and the risk of disease transmission [7,8]. The emergence of bone tissue engineering (such as biological scaffolding) provides a feasible method to overcome these limitations and has, thus, attracted the attention of many researchers in recent years [9,10,11]. Previous evidence shows that biological scaffolds can support the proliferation and metastasis of osteocytes [12,13,14], so they have the potential to meet the above requirements. In addition, scaffolds generally have good biocompatibility and can have a variety of different properties depending on their design, assembly, and surface modification; they have, therefore, become a focus in medical research [15,16,17]. Materials with excellent properties, such as polymeric hydrogels and electrospun fibers, are being increasingly used in bone tissue engineering with the continuous development of various scaffolds. Moreover, it is common to replicate natural bones by modifying ceramic compositions (e.g., hydroxyapatite) [18]. At present, many studies have used magnetic compositions as one of the scaffold materials for bone tissue engineering, and the results show that this material can promote osteoblast adhesion and increase alkaline phosphatase activity [3,19,20,21,22]. Due to the advantages of their special magnetic properties and the easy modification of their surface characteristics, superparamagnetic iron oxide nanoparticles (SPIONs) have achieved fruitful results in the fields of biosensors, non-invasive cell tracking, immune detection, drug delivery, and tumor diagnosis and treatment [23,24,25,26,27,28,29,30]. It is worth mentioning that SPIONs are often used as magnetic resonance imaging (MRI) probes for the imaging, detection, and diagnosis of bone-related diseases and can help to image early bone turnover changes such as osteoarthritis [31,32,33,34]. Due to the important properties of SPIONs, such as their good biocompatibility and superparamagnetic behavior, they have become an excellent choice for magnetic thermo-therapy and as a magnetic resonance imaging contrast agent [27,35,36,37,38,39,40]. Magnetic scaffolds can promote bone repair and regeneration via magnetic force, attracting and absorbing growth factors, stem cells, or other biological agents that combine with magnetic ions [20,41,42]. Because SPIONs play an important role in bone regeneration, bone scaffolds combined with SPIONs are a better choice for bone repair, especially when external magnetic stimulation is accompanied by external magnetic stimulation [43,44]. Natural bone tissue has complex hierarchical structures with varying scales of length and width composed of trabecular bone, haversian canals, osteons, lamellae, fibrillar collagen, minerals, collagen, and so on [13,45]. The mechanical properties of natural bones are different in different body parts. However, the longitudinal direction of compact bone is stronger than its transverse direction in bone tissue [13]. Moreover, natural bone contains cells, extracellular matrices, and bound minerals [13]. A scaffold containing SPIONs has structural features and functions close to those of natural bone, which can provide the scaffold with good biocompatibility, stiffness, and mechanical properties [18,43,44]. SPIONs can also be arranged in the scaffolds to replicate the anisotropic architectures of natural bones or for the magnetomechanical stimulation of engineered scaffolds during the maturation process, which has been demonstrated as essential in replicating native dynamic cellular environments [18]. SPIONs can make use of the superparamagnetism of magnetic compositions in cell microenvironments; enhance the osteogenesis and angiogenesis of scaffolds; and promote cell attachment, proliferation, and differentiation [18,43,44]. However, the shape and size of SPIONs can affect their application in bone tissue engineering [46,47]. Only SPIONs with uniform surfaces and sizes less than 20 nm are considered to exhibit superparamagnetic behavior; that is, becoming permanently magnetized by an external magnetic field [27,48]. In general, small SPIONs (<10 nm) show a rapid metabolism while larger ones (>200 nm) show a slow metabolism [49]. Moreover, the crystallinity and magnetic properties of SPIONs strongly influence their bioeffects in vivo [47]. Therefore, mastering the properties of SPIONs such as shape, size, concentration, and crystallinity can improve their combination with scaffolds in bone tissue engineering. In addition, it is important to design and synthesize magnetic nanostructures with a high magnetic response (large magnetization value). In this way, the fabricated scaffolds can be provided with a magnetic response by incorporating low amounts of magnetic material and then remotely manipulated by applying low-intensity magnetic fields, which could minimize the toxicity/safety risks associated with these factors and, in turn, increase the application potential of magnetic scaffolds [50]. Therefore, magnetic compositions with this desired high magnetic response can be obtained by controlling the size, morphology, composition, structure, and other factors of the structure [51]. This paper reviews several methods used for the synthesis of magnetic biological scaffolds via the biological assembly of SPIONs (see Table 1) and further discusses the application of these magnetic biological scaffolds in bone tissue engineering.

## 2. Biological Scaffolds Assembled with Magnetic Iron Oxide Compositions

Magnetic scaffolding is a biomaterial that has been widely studied in recent years; mainly those with magnetic and biological properties. Magnetic biomaterials also have great application potential in the field of bone tissue engineering. At present, the most commonly used magnetic particles in magnetic biomaterials are iron powder, iron trioxide, iron tetroxide, and so on [20,91]. The commonly used synthesis methods are freeze-drying, electrospinning, three-dimensional (3D) printing, chemical synthesis, and other methods (Figure 1).

### 2.1. Freeze-Drying

Freeze-drying is a technique for creating bioactive scaffolds with porous structures, generally employed to generate planar 3D geometric scaffolds. In the field of tissue engineering, biological matrix scaffolds provide a transitional framework to promote the development of new tissues, and the effects of physical and biological characteristics of scaffolds on the development of new tissues have been widely recognized. Freeze-drying can synthesize scaffold materials with uniform pore structures and a certain controllability, so it is often used as a method for the synthesis of biological scaffolds [92,93,94]. Bone is composed of hard minerals and flexible biopolymers to form a composite material with high strength and toughness [95]. Frank et al. [52] assembled a porous ceramic that can simulate sponge bone using Al_2_O_3_ particles magnetized on the surface (Figure 2). This kind of bioceramic is cast by the magnetic freezing method and has a multiaxially strengthened porous structure. Over the whole freezing process, the applied magnetic field can induce a specific arrangement of the particle chain and layered structure between the growing ice crystals. It can also control the electrostatic adsorption of SPIONs on the Al_2_O_3_ particles. This process can align the layered wall as desired and enhance the transverse stiffness, thus providing a method to solve the problem of layered wall misalignment caused by particle aggregation in the assembly process of porous scaffolds. Yang and his collaborators prepared collagen/cellulose nanocrystalline (COL/CNC) scaffolds using glutaraldehyde as the cross-linking agent and the conventional freeze-drying method [53]. Subsequently, a new cartilage-induced, small non-protein molecule—kartogenin (KGN)—was bound to the surface of the ultra-small SPIONs in a one-step method. The KGN was then compounded into cross-linked bioactive COL/CNC scaffolds to create a suitable microenvironment for the growth and differentiation of bone marrow mesenchymal stem cells (BMSCs), thus promoting the formation of chondrocytes. Among them, a SPION can be used not only as a carrier of small molecular drugs but also as an MRI contrast agent to non-invasively monitor the degradation of scaffolds and the self-repairing ability of cartilage. It is worth mentioning that the experimental results show that KGN can be continuously and steadily released from the composite scaffold and can promote the differentiation of BMSCs into chondrocytes. Kevin S. Tang et al. synthesized a magnetic poly (D,L-lactide-co-glycolide) (PLGA) scaffold via the freeze-drying method for magnetic cell labeling that can sufficiently enhance the redispersion of a PLGA-encapsulated iron oxide composition in water to enable single-cell detection via MRI [54]. Similarly, due to the microstructure of natural bone, the design and manufacturing of porous ceramic nanoscaffolds contained within the layers of a natural polymer could produce good scaffolds for bone tissue engineering. Amirsalar’s group fabricated multi-component porous magnetic scaffolds via the freeze-drying technique with good porosity and structural similarity to the natural bone that can be used in the treatment of bone cancer [55]. Xu and co-workers synthesized silk fibroin/hydroxyapatite (SF/HA) scaffolds using the freeze-drying method and combined them with ultramicro SPIONs [56]. SPION-labeled scaffolds are 3D structures with good porosity and mechanical properties and good thermal stability when used in bone repair, which can promote the adhesion and growth of BMSCs, thus promoting osteogenesis by increasing the activity of alkaline phosphatase (ALP) and upregulating the gene levels of osteoblasts. In addition, compared with pure SF and SF/HA stents, SPION-doped stents showed better thermal stability. These results show that biological scaffolds have good mechanical properties and thermal stability and are very suitable for application in bone tissue engineering. Díaz also used the freeze-drying method to prepare polycaprolactone/hydroxyapatite (PCL/HA) magnetic composite scaffolds with different compositions [57]. Magnetic measurements revealed the interaction between SPIONs. In addition, the biological scaffolds assembled by PCL and SPIONs can provide good matrix conditions for the migration, adhesion, and odontogenic differentiation of human dental pulp stem cells (DPSCs) [58]. Kim and co-workers are also interested in the biological scaffold assembled by PCL and SPIONs [59]. They also assembled a biological scaffold via the freeze-drying method and studied the combined effect of an external static magnetic field (SMF) and this magnetic composite scaffold on osteoblast function and bone formation. The magnetic scaffolds in this study contain up to 10% SPIONs and are dispersed in the PCL network to form a highly porous structure with the same characteristics as typical nanocomposite porous scaffolds. Due to the superparamagnetism of SPIONs, the nanocomposite scaffolds showed excellent magnetic properties. The results show that the combination of an external SMF and an internal magnetic field (magnetic composite scaffold) is a promising tool for bone regeneration engineering.

### 2.2. Electrospinning

Electrospinning is a powerful, quite simple, and widely used process. Electrospinning involves a jet erupting from the tip of a spinneret to produce fibers with diameters as low as the submicron or nanometer scales [96,97,98]. By optimizing the preparation process, these fibers can easily achieve minimal diameters, high porosities, a network geometry, controllable coverage, and controllable density [99,100]. A basic electrospinning device comprises four main parts: a high-voltage source; a syringe pump propulsion system; a spinneret; and a collector (Figure 3) [96,101]. The main principle of electrospinning technology is that the electrostatic force is caused by the electric field generated by the high-voltage device. When the electrostatic force overcomes the surface tension of the solution ejected from the syringe needle, the droplets formed by the solution at the outlet of the needle will form a Taylor cone, and the trickle of the solution will be ejected from the outlet of the needle. In this process, the solution is gradually stretched and refined, the solvent evaporates, and the dried polymer is arranged on the collector to form nanofibers [102]. Electrospinning uses a high-voltage power supply to create a large potential difference between the grounded “collector” structure and the polymer solution or melt, and the melt is transported at a constant speed through a small hole (such as a blunt-end needle) [103]. With the increase in voltage, the polymer fluid becomes charged, and the electrostatic repulsion is directly opposite to the surface tension, causing the usually spherical droplets at the hole to expand into a cone [102]. In the process of electrospinning, the applied voltage, the distance between the spinneret and the collector, the forward speed of the solution, the temperature and humidity, the relative molecular weight of the polymer, the polymer concentration, the surface tension of the polymer solution, and the conductivity of the polymer solution will affect fiber formation [102,104,105,106,107]. The impacts of these elements are described as follows. In terms of the applied voltage, increasing the voltage can make the spinning process easier. When the applied voltage increases, the fiber diameter will gradually decrease [108]. In terms of the distance between the spinneret and the collector, the receiving distance increases and the fiber diameter decreases; as the receiving distance decreases, the solvent cannot be completely volatilized in time, resulting in an uneven distribution of the electrospun fiber surface [108]. Retaining a moderate forward solution speed is beneficial to the formation of nanofibers. In a moderate range, with the increase in the forward speed of the solution, the diameter of the obtained nanofibers also increases [109]. Increases in temperature lead to increases in fiber diameter, and increases in humidity lead to decreases in the solvent volatilization rate [110,111]. When the relative molecular weight of the polymer is low, nano-scale fiber materials cannot be formed [112]. When the polymer concentration of the solution is too high, the viscosity of the solution will increase, which will block the needle of the syringe and prevent normal spinning from occurring [108,109]. In the process of electrospinning, if the surface tension of the charged droplets is too high and the force of the electrostatic field is less than the surface tension, the fibers will deform into droplets [113]. Increasing the conductivity of the polymer solution will accelerate the stretching of the solution, forming finer nanofibers [114]. Nanofiber scaffolds constructed by electrospinning have been shown to provide a better environment for tissue development because the 3D fiber matrix provides a structure similar to a 3D fiber network of collagen and elastin [101]. Wei and his collaborators obtained biodegradable magnetic nanofiber membranes by electrospinning SPIONs with chitosan (CS) and poly vinyl alcohol (PVA) [60]. The results show that the surface of the magnetic nanofiber film is uniform, smooth, and continuous, and the average fiber diameter is 230–380 nm. When the concentration of SPIONs is 4.5 wt% and the voltage is 20 kV, the porosity of the magnetic nanofiber membrane can reach 83.9–85.1%. With the increase in loaded SPIONs, the cell saturation magnetization, cell adhesion, and cell proliferation increased, indicating that the cell function on magnetic electrospun nanofiber membranes might be further regulated by controlling the amount of loaded SPIONs. Lee et al. prepared highly oriented fiber bundles composed of PLGA/SPION composite fibers via electrospinning [61]. Due to the unique properties of SPIONs, the prepared fiber bundles have superparamagnetism and there is no hysteresis. It is found that the surface morphology of the electrospun fiber bundle can spontaneously induce cell arrangement and form cell rods. After the treatment of the differentiation medium, the C2C12 myoblasts growing on the fiber bundles could fuse and differentiate into multinucleated myotubes. In addition, under the condition of an external magnetic field, the cell rods can self-assemble into 3D cell-dense tissue with a highly oriented structure. Yang’s group prepared poly-L-lactic acid (PLLA) and SPION composite nanofibers by electrostatic spinning to further verify the effect of magnetic matrix materials on the osteogenic differentiation of osteoblasts [62]. It was found that the involvement of SPIONs promoted osteoblast cell proliferation and osteogenic differentiation regardless of the presence of an external SMF. Further experiments revealed that the combination of magnetic nanofibers and an external SMF could further accelerate the biological behavior of osteoblasts. This reminds us that the repair process of osteogenic differentiation can be accelerated by adjusting the content of the SPIONs and applying an external SMF. As shown in Figure 4a, Meng et al. carried out a similar study, innovatively adding HA to magnetic nanofiber composite scaffolds [63]. The scaffold was implanted into the rabbit lumbar transverse defect model. With the help of an external SMF, the magnetic biological scaffold produced a large amount of micromagnetic force, which continuously stimulated osteoblast proliferation and the secretion of new extracellular matrix (ECM), thus promoting the formation and reconstruction of rabbit bone defect tissue. Similarly, Xu and co-workers found that fibrous SPION composite membranes prepared by electrostatic spinning were able to increase the proliferation rate and differentiation of osteoblasts in a culture medium under the presence of an external SMF [64]. In addition, Daňková’s group found that their SPION magnetic bioscaffold prepared by electrostatic spinning also enhanced the cell adhesion and proliferation of BMSCs, thus further supporting osteogenic differentiation [65]. Fiorilli and co-workers fabricated nanostructured magnetic scaffolds through the incorporation of SPIONs into a collagen scaffold during the electrospinning process to study its influence on cell activities in bone regeneration [66]. The magnetic properties of the SPIONs were preserved after their incorporation into the polymeric fibers. The scaffold improved the viability, adhesion, and proliferation of both pre-osteoblastic cells and human bone-marrow-derived mesenchymal stem cells (hBM-MSCs) and could, thus, serve as a potential platform for bone tissue regeneration. Mahsa Khalili et al. fabricated magnetic PCL nanofibers by incorporating third-generation dendrimer-modified SPIONs (G3–SPIONs) in the electrospinning process to further study the effect of magnetic scaffolds and pulsed electromagnetic fields (PEMF) on osteogenic potential [67]. The magnetic G3–SPION–PCL improved the growth and proliferation of stem cells, and increased the osteogenic differentiation of adipocyte-derived mesenchymal stem cells (ADMSCs) under a pulsed electromagnetic field, which is a promising magnetic scaffold for bone regeneration.

### 2.3. Three-Dimensional Printing

Many studies have shown that when cells grow on two-dimensional (2D) materials and in three-dimensional (3D) scaffolds, the cellular characteristics expressed—from function to morphogenesis—differ to varying degrees [115]. Three-dimensional scaffolds are closer to the natural internal environment of simulated cells, and it is relatively difficult to control the pore connectivity, pore size, and overall porosity of the scaffolds using other methods. Therefore, 3D printing technology has been used to overcome these problems and to prepare ideal scaffolds for bone tissue engineering [116,117,118]. Based on this, as shown in Figure 4b, some scholars have fabricated SPION magnetic biological scaffolds using 3D printing technology and applied them to magnetostatic-driven tissue engineering [68]. The related research results show that when the scaffold is labeled with MSCs and implanted subcutaneously, under the action of an external SMF, some of the tissues demonstrate osteogenesis and cartilage differentiation of stem cells in vivo. At the same time, it was found that MSCs expressed both bone- and cartilage-specific markers, which indicates that SPION magnetic biological scaffolds under magnetic stimulation could guide MSCs to differentiate into endochondral ossification. Zhang et al. [69] prepared 3D magnetic composite scaffolds containing SPION/PCL/mesoporous bioactive glass (MBG) using 3D printing technology and systematically studied the potential of these scaffolds in bone tissue engineering. The results show that the magnetic scaffold has a regular and uniform square macroporous structure, with a pore diameter of about 3.5 nm and a porosity as high as 60%. It is well known that a pore size greater than 100 μm can lead to cell inoculation, inward tissue growth, and angiogenesis. Nanopores within the micropore (<2 nm) or mesopore (2–50 nm) ranges allow for the transport of any nutrients, waste removal, and signal molecules, which promotes cell adhesion and the adsorption of biological agents. Therefore, 3D-printed SPION magnetic biological scaffolds have ideal graded pore structures and can be used for bone regeneration. Furthermore, the addition of SPIONs did not affect the apatite mineralization ability of the scaffold, but had excellent magnetic heating ability, which significantly stimulated the proliferation of BMSCs, alkaline phosphatase activity, and osteogenesis-related gene expression. At present, 3D bioprinting technology is attracting significant attention in the field of bone tissue engineering, and cells are being incorporated into bio-ink before manufacturing to produce scaffolds loaded with cells [119]. As an example of such a system, a magnetic bio-ink based on alginate and methylcellulose with incorporated magnetite microparticles was produced and shown to be highly compatible with the encapsulation of the human mesenchymal stem cell line (hMSC), thus promoting the development of 3D bioprinting technology [70]. In order to overcome the challenge of regenerating large bone fractures, Serpooshan’s group developed a 3D bioprinted scaffold with enhanced bacteriostatic properties and a highly porous structure [71]. SPIONs were incorporated into the hyperelastic bone (HB) scaffold via the 3D bioprinting technique, which enhanced the bacteriostatic properties of the produced bone grafts. The regenerative effect of the 3D scaffold on large, non-healing bone fractures was evaluated. The scaffold was implanted into a rat femoral defect model and showed a remarkable regeneration effect within two weeks.

### 2.4. Chemical Synthesis

Chemical synthesis is a commonly used method in the assembly of biological scaffolds. Chemical synthesis methods include chemical cross-linking to form hydrogels [41,120], the direct formation of insoluble solid substances [121], and further annealing treatment [72]. Combinations of SPIONs and biological scaffolds are mainly divided into two cases: one case is that, after the biological scaffolds are assembled, SPIONs are integrated into the scaffolds by impregnation, infiltration, or adsorption; the other case is that SPIONs are directly mixed with the matrix components of the biological scaffolds during the assembly process, which is used for the biological assembly of the scaffolds. Based on these two combination methods, the latest progress in the assembly of SPION magnetic biological scaffolds by chemical synthesis is described below.

Silk fibroin (SF), a protein obtained from silkworms, has excellent biocompatibility and was approved as a biomaterial by the FDA in 1993. In recent years, it has been widely used in the research on biological scaffolds [122]. Dediu and co-workers [73] assembled biological scaffolds using SF as the matrix. This approach introduced magnetic composition into SF scaffolds by the impregnation coating method and obtained magnetic SF scaffolds with different magnetization. Similarly, Eugenia Tanasa et al. prepared silk fibroin scaffolds decorated with magnetic composition to further study the impact of the magnetic field on preosteoblasts [74]. The results showed that the cellular proliferation of preosteoblasts increased under the magnetic field. Due to the presence of SPIONs, the magnetic scaffolds showed excellent hyperthermia properties under alternating magnetic fields and were able to raise the temperature to 8 °C in about 100 s. Moreover, the scaffold had good biocompatibility and improved the adhesion and colonization of osteoblasts. Recently, a gelatin sponge (GS) biological scaffold loaded with SPIONs was implanted into the incisor alveoli of rats for the first time, showing the scaffold’s effect on enhancing bone regeneration [31]. It has to be stated that this biological scaffold achieves a greater degree of new bone formation without the application of an external magnetic field, and the new bone formation is consistent with the degradation of the scaffold. In addition, this new method can be used to visually monitor bone repair in vivo in a non-invasive manner through MRI. Wu’s group assembled HA scaffolds and combined SPIONs with the impregnation method to study the effect of magnetism on bone repair and the interaction between magnetic scaffolds and external magnetic fields [72]. The results show that, compared with traditional HA scaffolds, magnetic HA scaffolds can accelerate bone formation and remodeling. The bone mineral density (BMD) four weeks after magnetic HA stent implantation was slightly higher than that after eight weeks of HA stent implantation. Notably, the presence of a magnetic field further accelerated the repair process: the bone mineral density at four weeks after magnetic HA stent implantation with an SMF was almost the same as that of 12 weeks after HA stent implantation without a magnetic field. This confirms that magnetic therapy based on magnetic HA scaffolds is a feasible bone repair strategy, especially in the early stage after transplantation.

Compared with the former binding method, another kind of SPION is more commonly used in biological scaffolds; that is, the SPIONs are mixed, cross-linked, or modified with the matrix materials of other scaffolds before assembly and then the entire biological scaffold is assembled. For example, SPION composite scaffolds with enhanced mechanical strength and versatility were successfully prepared by blending and reassembly [75]. The results showed that the inclusion of SPIONs did not significantly change the porosity or pore size of the scaffold but it did improve the compressive strength of the scaffold. What is important is that the SPION composite scaffolds show good biological activity and drug-release characteristics. At the same time, due to its intrinsic magnetic response, the composite scaffold can generate heat in the alternating magnetic field and increase the temperature of the surrounding environment, which is helpful in its application in bone tissue engineering. In addition, a new type of magnetic scaffold can be developed by mixing SPIONs into calcium phosphate cement (CPC) [76]. The experimental results show that the addition of SPIONs improves the characteristics of the CPC, including better wettability, greater protein adsorption, and greater cell adhesion and diffusion. Osteogenic differentiation was promoted by adding SPIONs and DPSCs into CPC, and the activity of ALP and the expression of osteogenic genes were also significantly increased. Compared with their absence, the addition of SPIONs increased bone matrix mineral synthesis by two to three times. Kim’s group carried out similar studies; they found that the most significant change is that BMSCs adhere to and spread well on CPC biological scaffolds containing SPIONs, and that cell proliferation lasts longer [77]. Coincidentally, some other scholars have a strong interest in promoting the osteogenic differentiation of DPSCs in SPION composite scaffolds. Based on the adjustment of the mechanical/interfacial properties of the used hydrogel, the multicellular sphere produced by the self-organization of the DPSCs showed a more significant osteogenic differentiation trend than the classical 2D-cultured DPSCs under the stimulation of SPIONs [78]. A tissue simulation particle composed of chondrocytes and hyaluronic acid/amphiphilic gelatin microcapsules was also proposed, and it was filled with SPIONs to simulate the extracellular matrix (ECM) environment of chondrocytes [79]. The preliminary results showed that, after adding SPIONs, the microcapsule shell not only showed good cell-guiding ability but also induced static magnetic field and magnetic source shear stress, showing the better growth and sequencing of chondrocytes. By synthesizing SPION scaffolds, Tampieri et al. [80] found that scaffolds can be reloaded by specific factors guided by an external SMF to help bone regeneration. They found that the combination of SPIONs and an SMF can be thought of as a cross-linking agent that increases the chemical, physical, and mechanical stability of the material and allows researchers to control the porosity network of the scaffold. A porous HA biological scaffold doped with SPIONs was used to repair severe femoral defects, and the potential of this magnetic biological scaffold for bone tissue repair was evaluated [81]. When implanted into a rabbit femur model, the magnetic porous HA scaffold showed excellent osteogenic ability compared to a commercial HA scaffold. Strong bone formation and proper osseointegration were observed four weeks after the operation and, although bone remodeling occurred at a later time point, the osseointegration of the stent was retained. At the same time, the fabrication and magnetothermal behavior of a novel SPION biological scaffold with linearly arranged colloidal components were reported [82]. To introduce the magnetic coupling between building blocks, field-oriented assembly is used to achieve the linear assembly of SPIONs and enhance the magnetic interaction between them. This novel SPION biological scaffold shows enhanced thermogenesis and controllable magnetocaloric behavior through experiments, depending on the direction of the external field relative to the assembly chain. Finally, the experimental results show that adjustable thermogenesis can be applied to the controlled drug release of magnetic hydrogels. As mentioned above, the SPION biological scaffolds were prepared using the freeze-drying assembly method; cellulose nanocrystals/dextran hydrogel were assembled via the chemical synthesis method; and SPIONs and KGN were mixed as fillers (Figure 4c) [83]. The magnetic biological hydrogel has good mechanical strength, and KGN can be released at a stable rate for a long time. Due to the presence of SPIONs, the hydrogel can also be used to carry out stable MRI imaging in vivo and in vitro. Surprisingly, this SPION hydrogel can recruit host BMSCs without cell transplantation, thus continuously promoting the regeneration of hyaline cartilage. A multifunctional MBG scaffold system has been developed for high-temperature and local drug delivery applications [84]. The related research results show that, after SPIONs are incorporated into MBG scaffolds, the shape of the mesopore changes from a straight channel to a curved fingerprint channel, and the magnetic properties of the MBG scaffolds can be adjusted by controlling the SPION content. In addition, the incorporation of SPIONs into MBG glass scaffolds could enhance the activity of mitochondria and the expression of bone-related genes (ALP and OCN) in the BMSCs attached to the scaffolds. Borosilicate bioactive glass scaffolds loaded with different levels (5–15 wt%) of SPIONs were constructed, and their performances in vitro and in vivo were evaluated [85]. The addition of SPIONs gave the scaffold excellent magnetism, controllable magnetocaloric properties, and higher mechanical capacity. With the increase in the amount of SPIONs in the scaffold, the ALP activity and osteogenic gene expression of BMSCs increased. When implanted into the skull defect sites of rats for eight weeks, the scaffold loaded with 15 wt% of SPIONs showed better and more significant bone-regeneration ability than the scaffold without SPIONs.

In addition, some scholars have compared the two methods of combining SPIONs with biological scaffolds. To evaluate the nanomechanical properties of newly formed bone tissue four weeks after the implantation of permanent magnets and magnetic stents in a rabbit femur, Bianchi designed two different assembly methods for SPIONs and biological NdFeB magnet/apatite/collagen scaffolds (i.e., SPIONs nucleated directly on collagen fiber during scaffold assembly and SPIONs infiltrated again after scaffold assembly) [86]. The results show that the regenerated bone tissue provided by the magnetic biological scaffold reinfiltrated by SPIONs after four weeks of implantation is closer to the mechanical properties of natural bone, which may be due to the better release performance of SPIONs combined with the scaffold in the later stage.

**Figure 4 materials-16-01429-f004:**
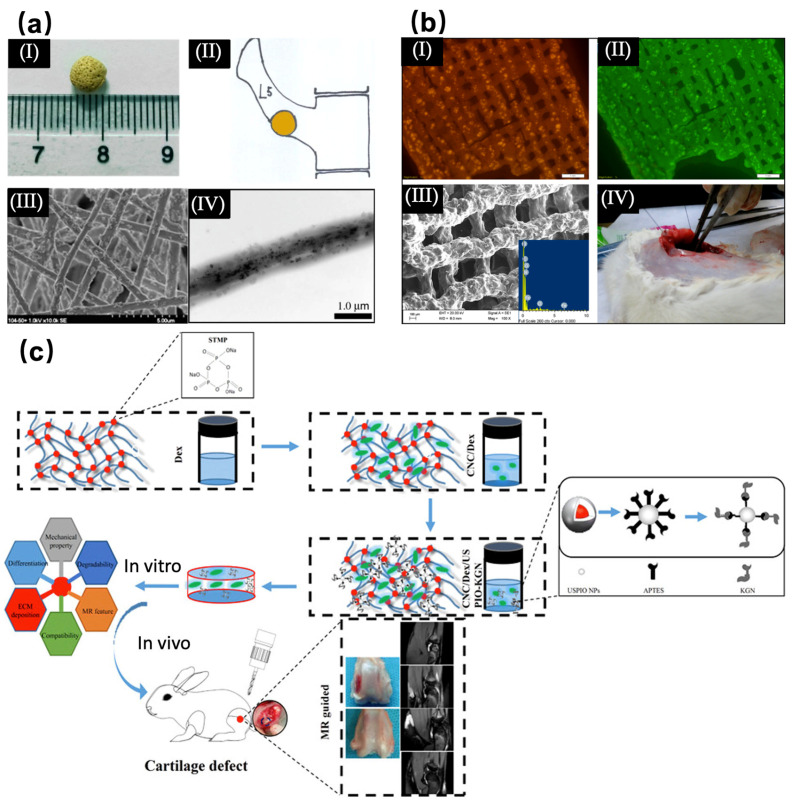
(**a**) Characterization of a superparamagnetic nanofiber scaffold. (I) Superparamagnetic nanofiber scaffold particles with a diameter of 5 mm. (II) Schematic diagram of the scaffold particles implanted in the defect of the L5 transverse process in rabbits. (III) SEM image of the scaffold. (IV) TEM image of the fibers in the scaffold. (Reproduced with permission from Ref [63], Copyright 2013, Springer Nature.) (**b**) Characterization of SPIONs–gelatin. (I) Visible image of SPIONs–gelatin (FeCD-Gel) 3D printing scaffold doped with carbon quantum dots. (II) Ultraviolet image of FeCD-Gel scaffold. (III) SEM image and EDAX analysis of FeCD-Gel scaffold. (IV) Image of FeCD-Gel scaffold implanted in a subcutaneous rat model. (Copyright 2019. Reproduced with permission from ref [68], Copyright 2019, ACS publication.) (**c**) Preparation of magnetic cellulose nanocrystal/dextran hydrogel and its application in artificial cartilage repair engineering. KGN is coupled on the surface of ultra-small superparamagnetic iron oxide. (Reproduced with permission from ref [83], Copyright 2019, ACS publication).

### 2.5. Other Methods

In addition to the more commonly used methods mentioned above, there are some rare methods used for the biological assembly of SPION magnetic scaffolds. Electrostatic layer-by-layer assembly (LBL) provides a multi-functional method to form multiple layers based on the alternating adsorption of charged polyelectrolytes, inorganic compositions, macromolecules, and even supramolecular systems on charged substrates [123]. Regardless of the size and topography of the substrate, the assembled multilayers are always uniformly formed on the substrate. Compared with the traditional immersion coating method, the advantage of LBL assembly is that it can refine the surface morphology and roughness through the assembly cycle [124]. Zhang and co-workers fabricated magnetic PLGA/PCL scaffolds using LBL assembly to further study the stimulation of cell growth [87]. SPIONs could enhance the hydrophilicity and increase the elastic modulus of the scaffold, which subsequently improved the osteogenic ability of stem cells, providing a novel method for the application of tissue engineering and regenerative medicine. When assembled on a porous material, the assembly is only formed on the solid surface without blocking the pores, which is of great significance. Tang successfully applied a silica composition to PCL fiber scaffolds using LBL assembly to improve the wettability and roughness of the fibers, thus enhancing the proliferation and adhesion of osteoblasts [125]. An external SMF is very beneficial to bone growth [126] as it can be considered to produce a local magnetic effect by providing a magnetic interface around the tissue defect. In view of this, a SPION magnetic biofilm scaffold was developed by adsorbing SPIONs on the surface of substrate through the LBL method [88]. Depending on the electrostatic interaction between the composition and polyelectrolytes, SPIONs are adsorbed on the glass slides to form bilayers. Because of this, the magnetic biofilm scaffold is very stable, even in complex media. The expression of the magnetic-sensitive protein shows that the assembly of the magnetic composition has a magnetic effect on cells, suggesting that it may be a promising scaffold interface that can combine its physical effects with its good biocompatibility, promoting the growth and differentiation of stem cells. In addition, the modification of SPIONs enhanced the mechanical properties of the interface and improved the biocompatibility of the scaffolds.

The evaporation-induced self-assembly (EISA) process is also commonly used to quickly produce patterned porous or nanocomposites in the form of thin films, fibers, or powders [127]. Zhu and his collaborators used EISA to assemble SPION scaffolds using polyurethane sponge and P123 as co-templates [89]. The results show that the bio-assembled SPION scaffold has a connected macroporous structure with a pore size range of 200–400 μm, a peak pore diameter of the mesoporous wall of about 3.34 nm, and a porosity as high as 86.4%. It is not difficult to understand that these macropores are conducive to cell proliferation, vascular growth, and internal mineralized bone formation. More importantly, the SPION magnetic scaffold possesses superparamagnetism and can generate heat under an alternating magnetic field, which has potential application value in hyperthermia. Moreover, Chen and co-workers constructed a AuNPs/SPIONs@ cobalt protoporphyrin IX scaffold using nano self-assembly. The scaffold shows excellent biocompatibility and magnetic manipulation ability, with a potential application for labeling mesenchymal stem cells (MSCs) and the potential for the development of bone tissue engineering [90]. 

## 3. Application of Magnetic Scaffold in Bone Repair and Cartilage Repair

### 3.1. Toxicity

SPIONs are widely used in bone tissue engineering; therefore, the toxicity of these material has aroused great concern. In recent years, there have been many studies on the toxicity of SPIONs but, according to the research, their toxicity is relatively small [27,128]. Although organic chemicals are used in the preparation process, the synthesized SPIONs showed good hydrophilicity and biocompatibility [47,129]. Moreover, the SIPONs prepared from the chemical co-precipitation method in water solution showed better hydrophilicity and biocompatibility than those prepared in organic solution [47]. For example, some SPIONs prepared by the classic chemical co-precipitation method have been approved by the FDA for clinical use [130]. It has been found that SPIONs generally accumulate at a high level in the kidneys and organs of the reticuloendothelial system, including the liver, spleen, and bone marrow [131]. A study showed that a dose of SPIONs exceeding 35 mg/kg will cause significant toxicity to the liver and kidneys, indicating that the toxicity is dose-dependent [132]. The metal materials in SPIONs cannot be cleared by the body because they are non-biodegradable particles [27]. Similarly, the long-term complete elimination of SPIONs is also uncertain. It has been found that the clearance of SPIONs obviously depends on the dose, and higher doses were proven to take longer to completely clear [44,133,134].

### 3.2. Bone Repair

In recent years, efforts to repair and treat bone fractures and defects have paid more attention to cell therapy. SPION magnetic scaffolds have good therapeutic effects and obvious advantages in these aspects, attracting numerous scholars to apply them in bone tissue engineering studies [44]. For example, a SPION magnetic scaffold with HA and collagen was assembled, and BMSCs were cultured on the scaffold in the reference Bock et al. [135]. The results showed that the scaffolds supported the proliferation of BMSCs well. Honda’s group conducted a similar experiment and reached the same conclusion [136]. Compared with the group treated without an SMF, the density of BMSCs on the SPION magnetic scaffold increased by three times after adding the SMF, and the levels of the two representative osteogenic markers—ALP and osteocalcin (OC)—were also significantly increased, indicating that the presence of an SMF is conducive to the induction of osteocyte formation. We assembled a SPION magnetic film scaffold using LBL assembly and explored the supporting effect of this kind of scaffold on BMSCs [88]. It was found that, after 15 days of culture, the expression of some proteins related to cell differentiation—including OC, osteopontin (OPN), dwarf-related transcription factor 2 (RUNX2), and bone morphogenetic protein 2 (BMP-2)—were increased. Among them, OC and OPN are indicators of osteoblast differentiation, while RUNX2 and BMP-2 are upstream and downstream proteins of osteogenic differentiation. We thus speculate that this SPION magnetic membrane scaffold may promote osteogenesis by upregulating the expression of these proteins. Some studies have confirmed that SPIONs promote the differentiation of BMSCs into osteoblasts through the TGF-β, PI3K-AKT, and calcium signaling pathways, which inhibits the differentiation of mononuclear bone marrow macrophages into osteoclasts through the TRAF6–CYLD–p62 signaling complex [137]. Some scientists implanted SPION magnetic scaffolds assembled by electrospinning into a model of a lumbar transverse defect in rabbits to verify the effect of bone repair. The results show that the addition of SPIONs to the biological scaffold resulted in the scaffold having a superparamagnetic response under the action of the static magnetic field, which greatly promotes the formation and reconstruction of rabbit bone tissue. After the scaffold was implanted in rabbits, it showed good compatibility with serum creatine kinase (CK), serum creatinine (Cr), glutamic pyruvic transaminase (ALT), and ALP within 110 days. Xia and co-workers implanted GS loaded with SPIONs as a scaffold material into the incisor sockets of rats, and the anterior alveolar was filled with a blank GS as a control [31]. At two weeks, the incisor fossa of tooth extraction was full of neovascularization, connective tissue, and some new bone. The activity of osteoblasts around the new bone increased, and the formation of blood vessels was accompanied by bone growth. Four weeks after the operation, the incisor fossa of the extracted tooth was partially repaired but the structure of the new bone was not as mature as the natural bone. In the absence of an external magnetic field, the mechanical stress signal produced at the beginning of the interaction between the cell membrane and the SPIONs may promote the differentiation of osteoblasts. Subsequently, this interaction may lead to extensive regulation of gene expression and activation of the classic mitogen-activated protein kinase signaling pathway [138]. Therefore, the downstream genes of the mitogen-activated protein kinase signaling pathway are regulated to promote osteogenic differentiation. 

Angiogenesis is very important in osteogenesis because oxygen supply is generally no further than 200 μm away from blood vessels; otherwise, without good blood supply, the cells will not survive, and the formation of new bone will be hindered. Therefore, efficient angiogenesis is very necessary for bone regeneration. The increase in new bone may be due to the synergistic effect of integrin, BMP, mitogen spark protein kinase (MAPK), and nuclear factor κ B (NF- κ B) signaling pathways in osteoblasts cultured with an SMF and magnetic scaffolds [59,139]. SPIONs can inhibit the differentiation of mononuclear macrophages into osteoclasts in bone marrow following phagocytosis by the reticuloendothelial system [140]. Mechanically, SPIONs trigger the upregulation of p62, which leads to the recruitment of CYLD and the enhancement of TRAF6 de-ubiquitin. The downstream activation of the NF-kappa B and MAPK signals is, therefore, weakened, resulting in a decrease in the expression of genes related to osteoclast formation [140]. A SPION magnetic biological scaffold developed by Ai and co-workers significantly prevented bone loss in ovariectomized mice, increased BMD by 9.4%, and led to the overexpression of osteoprotegerin (OPG), CSF2, CCL2, and other cytokines responsible for maintaining the balance of bone remodeling [137]. Shen’s research shows that, once SPIONs are incorporated into the MBG/PCL scaffold, the micro-environment in the scaffold hole or on the surface of the scaffold is composed of a large number of tiny magnetic fields. The overall effect may be enhanced with the increase in the number of SPIONs, which in turn affects the ion channels on the cell membrane, showing the effect of bone induction under the action of the magnetic field [141].

To summarize, SPION magnetic biological scaffolds rely either on their own mechanical stress or on the magnetic effects produced by an SMF to promote the migration, adhesion, proliferation, and differentiation of BMSCs, DPSCs, or other cells related to osteogenesis. They also rely on these to upregulate the expression of some cytokines, bone formation-related genes, and bone formation-related proteins to achieve bone tissue and cell repair, regeneration, renewal, and so on.

### 3.3. Cartilage Repair

Compared with bone repair, cartilage repair is a thornier problem in clinical research. Because articular cartilage lacks blood vessels and nerves, and chondrocytes are wrapped in the cartilage matrix, it is, therefore, not easy for cells to migrate to the injured area to participate in repair. Based on this, scientists began to explore the use of biological scaffolds for cartilage repair. For example, a CNC-enhanced dextran (Dex) scaffold was designed in which water-soluble linear Dex and sodium trimetaphosphate were cross-linked, and SPIONs were mixed into the scaffold [83]. Dex has been studied as a potential biological scaffold for repairing cartilage because of its good biocompatibility, non-toxicity, and non-immunogenicity. Due to its lack of healing ability, cartilage regeneration is a long process. The degradation rate of the nutrient matrix must match the regeneration rate of cartilage. It was found that the expression of BMSC cartilage markers in SPION magnetic scaffolds was significantly higher than that in KGN-free hydrogels after 14 days of induction. For example, compared with the biological scaffold without SPIONs, the mRNA expression of the proteoglycan, COL1A2, and SOX9 of cartilage in the SPION magnetic biological scaffold increased by about 214.2 ± 39.5, 308.2 ± 31.5, and 197.0 ± 10.3, respectively, on the 14th day. This suggests that SPION magnetic scaffolds promote the phenotype of hyaline cartilage rather than fibrocartilage and maintain the phenotype of chondrocytes in the matrix. In addition, SPION magnetic biological scaffolds had the best effect on cartilage regeneration, and most of the defects were repaired after the sixth week. At the 12th week, a regenerated cartilage-like tissue with a smooth surface and borderless fusion with the adjacent host cartilage was observed, indicating that the degeneration of normal cartilage caused by certain defects was inhibited. In SPION magnetic biological scaffolds, the continuously generated new cartilage is similar to normal cartilage and fuses well with it. What can be explained by the above is that the KGN released in SPION magnetic scaffolds promotes the differentiation of BMSCs and even shows cartilage protection. The SPION magnetic scaffolds can recruit host BMSCs without cell transplantation, thus promoting the continuous regeneration of hyaline cartilage. Furthermore, the inflammatory reaction of the cartilage defect after implantation was investigated, and it was found that there were no obvious signs of inflammation in all groups. The Il-1 and TNF-α levels initially increased during the repair period and were then maintained at a very low level, indicating that the implantation of the SPION magnetic biological scaffold will not cause inflammation or rejection. Guo’s group also found, through experiments, that the continuous release of KGN may promote the growth of cartilage cells better and can summon host endogenous cells—including BMSCs—to the defect site, promoting their proliferation and differentiation [53]. In addition, if SPION magnetic scaffolds are implanted into cartilage defects, the degradation and regeneration of the cartilage in vivo can be monitored by MRI. A tissue simulation sphere composed of chondrocytes and hyaluronic acid graft/amphiphilic gelatin microcapsules was designed, and SPIONs were wrapped in it [79]. Studies have shown that the SPION magnetic microcapsules have good structural stability and can maintain good cell compatibility and vitality. When an external SMF and magnetic shear stress were continuously applied to SPION magnetic microcapsules for five days, the expressions of COLI, COLII, and SOX9 genes were significantly increased compared with the control group. Other scholars have used PVA to coat SPIONs. Related studies have shown that this SPION magnetic biological scaffold is a promising delivery system for magnetic drug targeting in synovial tissue because they are absorbed both in vitro and in vivo [142]. Due to the highly anisotropic tissue of cartilage, magnetic scaffolds can be incorporated into the biomaterials and then remotely arranged in a controlled manner, through the application of external magnetic fields, to replicate the anisotropic architecture of native cartilages in the scaffolds [143,144]. For example, magnetic collagen agarose hydrogels were used to produce magnetic scaffolds by 3D bioprinting technology [145]. The magnetic scaffold can control the alignment of collagen fibers. The presence of magnetic particles and magnetic field triggered the alignment of collagen fibers in the desired direction, which obtained a bioprinted scaffold with alternating layers of aligned and random fibers. Based on this anisotropic structure, the magnetic composite scaffold enhanced mechanical stiffness and expressed more collagen II compared with single-layer materials, improving the potential of bioprinted scaffolds in cartilage regeneration. In summary, SPION magnetic biological scaffolds can achieve cartilage repair and regeneration by increasing the expression of cartilage-osteogenesis-related genes, summoning BMSCs to the designated site and promoting their expression, and causing less inflammatory reactions, in addition to other bone components and their own magnetocaloric effect.

## 4. Conclusions

Increasingly, scientific studies have shown that SPIONs have great potential in bone tissue engineering. The research and development of SPION magnetic biological scaffolds outlined in this paper provides new methods for the development and treatment of bone regeneration. This paper introduces several assembly methods of SPION magnetic biological scaffolds, including freeze-drying, electrospinning, 3D printing, chemical synthesis, and other rarer methods, and summarizes their applications in bone tissue engineering. SPION biological scaffolds are generally implanted into the target bone defects without adding an SMF or under the action of one to stimulate bone tissue repair, regeneration, healing, and so on. When exposed to SPION magnetic biological scaffolds, osteoblasts, BMSCs, and DPSCs showed active migration, proliferation, and differentiation, while osteoclasts showed abnormal behavior, which may be attributed to the effect of the micromagnetic field, the magnetic mechanical stimulation, and the increase in the intracellular SPION level in the presence of an SMF [91]. At the same time, the expression levels of some genes and cellular molecules related to osteogenesis are significantly increased, which further promotes the repair of bone and cartilage defects. It is clear that the application of superparamagnetic response scaffolds with external magnetic fields provides a new strategy for scaffold-guided bone repair, and the combined application of SPIONs and SMFs can become a non-invasive and convenient treatment to promote bone regeneration [91].

Furthermore, although SPIONs have been approved for clinical use, the vast majority of SPION magnetic scaffolds lack adequate safety assessment. Therefore, the role of SPION magnetic scaffolds in vivo requires long-term studies to further determine their possible adverse effects on organisms [20,91]. Magnetic scaffolds may produce uncontrolled aggregation in the biological environment or release metal ions that are potentially toxic to cells [50]. Therefore, designing highly monodisperse functionalized SPIONs can prevent uncontrolled aggregation from occurring. Moreover, magnetic scaffolds with high magnetic energy contain low amounts of magnetic material and require lower intensities of magnetic radiation for remote control, thus reducing the related toxicity [50]. However, there are still many challenges related to the development of magnetic scaffolds in the bone tissue engineering field. This generally requires the use of novel processing techniques. For example, magnetically assisted 3D bioprinting techniques can be exploited to design magnetically responsive, cell-laden scaffolds with impressive control over their resolution and shape fidelity, broadening the available design space of hybrid magnetic composites [119]. Magneto-ceramic compositions are also promising materials with high magnetic properties that could promote further development in bone tissue engineering [146]. To summarize, SPION magnetic biological scaffolds have obvious advantages and great therapeutic ability, endowing them with very broad application prospects in bone tissue engineering.

## Figures and Tables

**Figure 1 materials-16-01429-f001:**
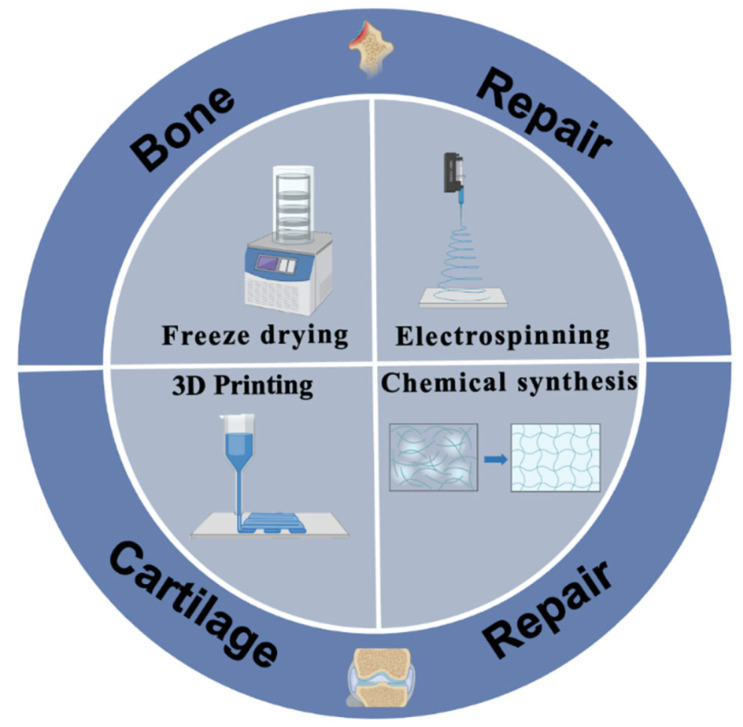
Schematic illustration of the fabrication of magnetic biological scaffolds and their biomedical applications.

**Figure 2 materials-16-01429-f002:**
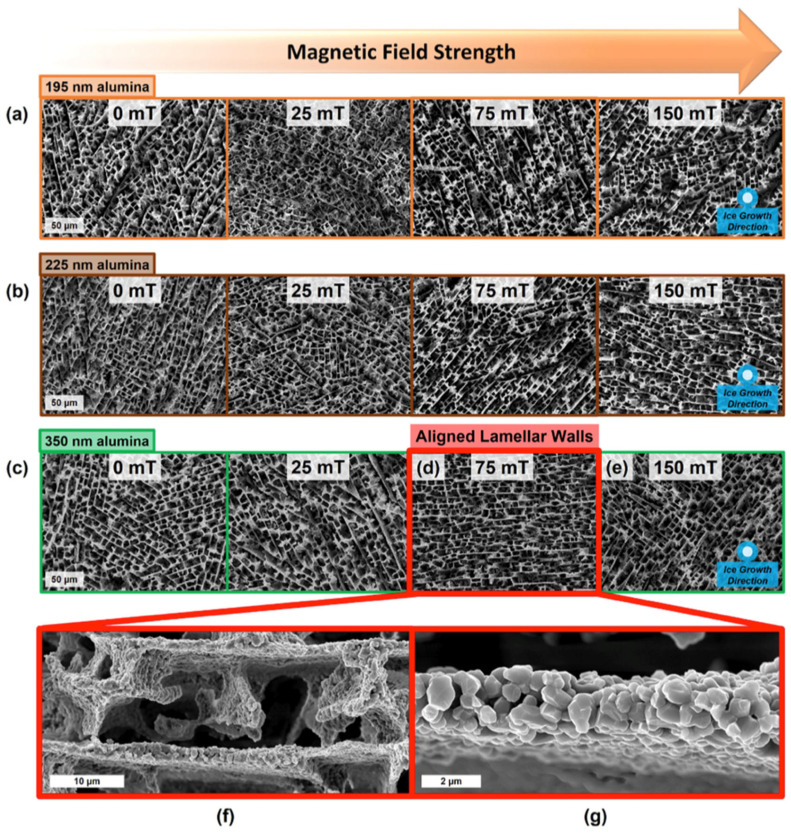
Scanning electron microscope photos of scaffolds with different sizes of magnetized alumina particles frozen-cast in 0, 25, 75, and 150 mT magnetic fields. (**a**) The 195 nm magnetized alumina particles did not produce lamellar wall alignment. (**b**) The 225 nm magnetized alumina particles exhibit limited alignment at 150 mT magnetic field. (**c**) Alignment of 350 nm magnetized alumina particles with lamellar wall, which was most evident at (**d**) 75 mT. (**e**) Alignment of 350 nm magnetized alumina particles was angled possibly due to flux field effects at 150 mT magnetic field. (**f**) The microscopic mineral bridge of the lamellar wall. (**g**) Magnetized alumina particles aligned within the lamellar wall along the magnetic field direction. (Reproduced with permission from Ref [52], Copyright 2017, Elsevier).

**Figure 3 materials-16-01429-f003:**
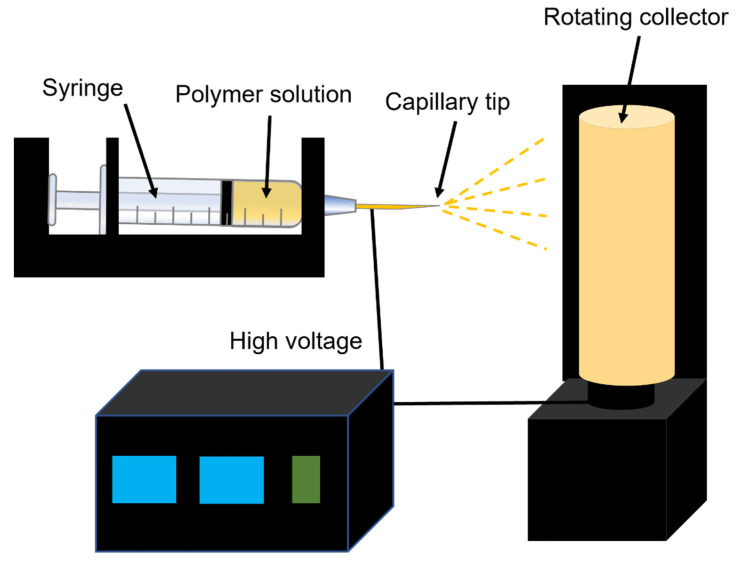
Schematic diagram showing the electrospinning device.

**Table 1 materials-16-01429-t001:** Biological scaffolds assembled with SPIONs and their applications in bone tissue engineering.

Methods of Preparation	Scaffold Material	NPs Composition	Applications	Ref.
Freeze-drying method	Alumina particles	Superparamagnetic Fe_3_O_4_	Stiff, porous scaffolds	[52]
	The cross-linking collagen/cellulose nanocrystals	The ultra-small super-paramagnetic iron-oxide (USPIO)	Cartilage regeneration	[53]
	PLGA	SPIONs	Cell labeling	[54]
	Gentamicin-gelatin-coated on porous ceramic	Magnetic composition	Bone cancer treatment	[55]
	Silk fibroin/hydroxyapatite	USPIO	Bone regeneration	[56]
	Polycaprolactone	FeHA nanoparticle	Bone tissue engineering	[57]
	Polycaprolactone	Magnetic composition	Migration and odontogenesis of human dental pulp cells	[58]
	Polycaprolactone	Magnetic composition	Osteoblastic differentiation and bone formation	[59]
Electrospinning method	Chitosan/poly vinyl alcohol	Fe_3_O_4_ composition	Bone regeneration	[60]
	Poly (L-lactide-co-glycolide)	SPIONs	The construction of a 3D cell-dense engineered tissue	[61]
	Poly (L-lactide)	Ferromagnetic Fe_3_O_4_ composition	Osteogenic differentiation	[62]
	Hydroxyapatite composition and poly lactide acid	Super-paramagnetic γ-Fe_2_O_3_ composition	Bone repair	[63]
	Poly lactide, hydroxyapatite	γ-Fe_2_O_3_ composition	The osteogenic responses of pre-osteoblast cells	[64]
	Poly-ε-caprolactone	Magnetic composition	Mesenchymal stem cell proliferation	[65]
	Collagen	SPIONs	Bone regeneration	[66]
	Polycaprolactone	SPIONs	Osteogenic differentiation	[67]
Three-dimensional printing technique	Carbon nanodots	SPIONs	Multimodal bioimaging and osteochondral tissue regeneration	[68]
	Mesoporous bioactive glass/polycaprolactone	Magnetic Fe_3_O_4_ composition	Bone regeneration, local anticancer drug delivery, and hyperthermia	[69]
	Alginate and methylcellulose	Magnetite composition	The encapsulation and cultivation of cell	[70]
	Hydroxyapatite	SPIONs	Bone regeneration	[71]
Chemical synthesis method	Hydroxyapatite	SPIONs	Bone repair	[72]
	Silk fibroin protein	Magnetic composition	Osteogenic cell differentiation	[73]
	Silk fibroin	Magnetic composition	Proliferation of cell	[74]
	Gelatin sponge	SPIONs	Bone regeneration and visual monitoring	[31]
	Mesoporous bioactive glass/carbon	Magnetic composition	Bone regeneration	[75]
	Calcium phosphate cement	γFe_2_O_3_ composition and αFe_2_O_3_ composition	Osteogenic differentiation	[76]
	Calcium phosphate cement	Magnetic composition	Bone regeneration	[77]
	The GelMA/PEGDA composite hydrogel	Magnetic iron oxide composition	Osteogenic/odontogenic differentiation of dental pulp stem cells	[78]
	Rabbit primary chondrocytes and hyaluronic acid-graft-amphiphilic gelatin microcapsules	SPIOs	Chondrogenic regeneration	[79]
	Hydroxyapatite/collagen	Magnetite composition	Bone regeneration	[80]
	Hydroxyapatite	Magnetite composition	Bone regeneration in a rabbit critical femoral defect	[81]
	Hydrogel	Magnetic composition	Controlled drug release	[82]
	Cellulose nanocrystal/dextran hydrogels	USPIO	Cartilage regeneration	[83]
	Mesoporous bioactive glass	Magnetic composition	Hyperthermic and local drug delivery applications	[84]
	Borosilicate bioactive glass	Fe_3_O_4_ magnetic composition	Bone regeneration	[85]
	NdFeB magnet + apatite/collagen	Magnetic composition	Bone regeneration	[86]
Layer-by-layer method	Poly(lactic-co-glycolic acid)/polycaprolactone (PLGA/PCL)	SPIONs	Osteogenesis of the stem cells	[87]
	Poly-D, L-lactic acid	γ-Fe_2_O_3_ composition	The growth and differentiation of primary bone marrow cells	[88]
Evaporation-induced self-assembly method	Polyurethane sponge and P123	Fe_3_O_4_ magnetic composition	Bone regeneration	[89]
	Cobalt protoporphyrin IX	AuNPs/SPIONs	Cell labeling	[90]

## Data Availability

Not applicable.

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
