# Peer review of "Biological Scaffolds Assembled with Magnetic Nanoparticles for Bone Tissue Engineering: A Review"

_materials, 2023, doi:10.3390/ma16041429_

Round 1
Reviewer 1 Report
The current work focuses on Biological scaffolds assembled with magnetic iron oxide nanoparticles for bone tissue engineering, but minor issues should be addressed. I believe that the material presented in the manuscript is interesting for specialists in magnetic materials for bone tissue engineering. It can be published in the journal after additions and revisions.
- Biological scaffolds assembled with magnetic iron oxide nanoparticles:
“Used synthesis methods are freeze-drying, electrospinning, three-dimensional (3D) printing, chemical synthesis, and other methods”. More details are required to show the properties of the advantages and disadvantages of each method.
- ”At present, the commonly used magnetic particles in magnetic biomaterials are iron powder, iron trioxide, iron tetroxide, and so on.” This sentence needs citation.
- For biomedical applications:
The toxicity and availability of the fabricated materials should be clear. A subsection related to toxicity should be inserted to clear this point.
- Recent citations on magnetic nanomaterial for biomedical applications should include supporting the review
https://doi.org/10.3390/ma15217823
https://doi.org/10.3390/ijms23031023
https://doi.org/10.3390/magnetochemistry7100133
- Summary and prospect:
There is a lack of critical assessments by the authors. The authors did not mention the research gap between the previously reported articles and the present situation. Authors should incorporate their views to mold the research in a new direction.
- References
Should have one system style
Author Response
Reviewer #1:
- Biological scaffolds assembled with magnetic iron oxide nanoparticles:
“Used synthesis methods are freeze-drying, electrospinning, three-dimensional (3D) printing, chemical synthesis, and other methods”. More details are required to show the properties of the advantages and disadvantages of each method.
[Response]: Thanks for the comments. We have added some details in freeze drying, electrospinning, three-dimensional (3D) printing, chemical synthesis and other methods to show the advantages and disadvantages of each method.
[Action taken]: In Page 6, the freezing drying method, line 25, we added the following description “Kevin S. Tang et al. synthesized a magnetic poly(D,L-lactide-co-glycolide) (PLGA) scaffold via the freeze-drying method for magnetic cell labeling that can sufficiently enhance the redispersion of a PLGA-encapsulated iron oxide composition in water to enable single-cell detection via MRI[54]. Similarly, due to the microstructure of natural bone, the design and manufacturing of porous ceramic nanoscaffolds contained within the layers of a natural polymer could produce good scaffolds for bone tissue engineering. Amirsalar's group fabricated multi-component porous magnetic scaffolds via the freeze-drying technique with good porosity and structural similarity to the natural bone that can be used in the treatment of bone cancer [55].”. In Page 9, the Electrospinning method, line 17, we added the following description “Fiorilli and co-workers fabricated nanostructured magnetic scaffolds through the incorporation of SPIONs into a collagen scaffold during the electrospinning process to study its influence on cell activities in bone regeneration[66]. The magnetic properties of the SPIONs were preserved after their incorporation into the polymeric fibers. The scaffold improved the viability, adhesion, and proliferation of both pre-osteoblastic cells and human bone-marrow-derived mesenchymal stem cells (hBM-MSCs) and could thus serve as a potential platform for bone tissue regeneration. Mahsa Khalili et al. fabricated magnetic PCL nanofibers by incorporating third-generation dendrimer-modified SPIONs (G3-SPIONs) in the electrospinning process to further study the effect of magnetic scaffolds and pulsed electromagnetic fields (PEMF) on osteogenic potential[67]. The magnetic G3-SPION-PCL improved the growth and proliferation of stem cells, as well as increased osteogenic differentiation of adipocyte-derived mesenchymal stem cells (ADMSCs) under a pulsed electromagnetic field, which is a promising magnetic scaffold for bone regeneration.”. In Page 10, the 3D printing method, line 29, we added the following description “At present, 3D bioprinting technology is attracting great attention in the field of bone tissue engineering, and cells are being incorporated into bio-ink before manufacturing to produce scaffolds loaded with cells[119]. As an example of such a system, a magnetic bio-ink based on alginate and methylcellulose with incorporated magnetite microparticles was produced and shown to be highly compatible with the encapsulation of human mesenchymal stem cell line (hMSC), thus promoting the development of 3D bioprinting technology[70]. In order to overcome the challenge of regenerating large bone fractures, Serpooshan's group developed a 3D bioprinted scaffold with enhanced bacteriostatic properties and a highly porous structure[71]. SPIONs were incorporated into the hyperelastic bone (HB) scaffold via the 3D bioprinting technique, which enhanced the bacteriostatic properties of the produced bone grafts. The regenerative effect of the 3D scaffold on large, non-healing bone fractures was evaluated. The scaffold was implanted into a rat femoral defect model and showed a remarkable regeneration effect within 2 weeks.”. In Page 11, the Chemical synthesis method, line 6, we added the following description “Similarly, Eugenia Tanasa et al. prepared silk fibroin scaffolds decorated with magnetic composition to further study the impact of the magnetic field on preosteoblasts[74]. The results showed that the cellular proliferation of preosteoblasts increased under the magnetic field.”. In Page 14, the Other methods, line 4, we added the following description “Zhang and co-workers fabricated magnetic PLGA/PCL scaffolds using layer-by-layer assembly to further study the stimulation of cell growth[87]. SPIONs could enhance the hydrophilicity and increase the elastic modulus of the scaffold, which subsequently improved the osteogenic ability of stem cells, providing a novel method for the application of tissue engineering and regenerative medicine.”. In Page 14, the Other methods, line 35, we added the following description “Moreover, Chen and co-workers constructed a AuNPs/SPIONs@ cobalt protoporphyrin IX scaffold using nano-self-assembly. The scaffold shows excellent biocompatibility and magnetic manipulation ability, with a potential application labeling mesenchymal stem cells (MSCs) and providing potential for the development of bone tissue engineering[90]. ”.
- Tang, K. S.; Hashmi, S. M.; Shapiro, E. M., The Effect of Cryoprotection on the Use of PLGA Encapsulated Iron Oxide Nanoparticles for Magnetic Cell Labeling. Nanotechnology 2013, 24 (12).
- Nassireslami, E.; Motififard, M.; Moghadas, B. K.; Hami, Z.; Jasemi, A.; Lachiyani, A.; Foroushani, R. S.; Saber-Samandari, S.; Khandan, A., Potential of Magnetite Nanoparticles with Biopolymers Loaded with Gentamicin Drug for Bone Cancer Treatment. Journal of Nanoanalysis 2021, 8 (3), 188-198.
- Estevez, M.; Montalbano, G.; Gallo-Cordova, A.; Ovejero, J. G.; Izquierdo-Barba, I.; Gonzalez, B.; Tomasina, C.; Moroni, L.; Vallet-Regi, M.; Vitale-Brovarone, C.; Fiorilli, S., Incorporation of Superparamagnetic Iron Oxide Nanoparticles into Collagen Formulation for 3D Electrospun Scaffolds. Nanomaterials 2022, 12 (2).
- Khalili, M.; Keshvari, H.; Imani, R.; Sohi, A. N.; Esmaeili, E.; Tajabadi, M., Study of Osteogenic Potential of Electrospun PCL Incorporated by Dendrimerized Superparamagnetic Nanoparticles as a Bone Tissue Engineering Scaffold. Polymers for Advanced Technologies 2022, 33 (3), 782-794.
- Spangenberg, J.; Kilian, D.; Czichy, C.; Ahlfeld, T.; Lode, A.; Gunther, S.; Odenbach, S.; Gelinsky, M., Bioprinting of Magnetically Deformable Scaffolds. Acs Biomaterials Science & Engineering 2021, 7 (2), 648-662.
- Shokouhimehr, M.; Theus, A. S.; Kamalakar, A.; Ning, L.; Cao, C.; Tomov, M. L.; Kaiser, J. M.; Goudy, S.; Willett, N. J.; Jang, H. W.; LaRock, C. N.; Hanna, P.; Lechtig, A.; Yousef, M.; Martins, J. D. S.; Nazarian, A.; Harris, M. B.; Mahmoudi, M.; Serpooshan, V., 3D Bioprinted Bacteriostatic Hyperelastic Bone Scaffold for Damage-Specific Bone Regeneration. Polymers 2021, 13 (7).
- Tanasa, E.; Zaharia, C.; Hudita, A.; Radu, I.-C.; Costache, M.; Galateanu, B., Impact of the Magnetic Field on 3T3-E1 Preosteoblasts Inside SMART Silk Fibroin-Based Scaffolds Decorated with Magnetic Nanoparticles. Materials Science and Engineering C-Materials for Biological Applications 2020, 110.
- Chen, H.; Sun, J.; Wang, Z.; Zhou, Y.; Lou, Z.; Chen, B.; Wang, P.; Guo, Z.; Tang, H.; Ma, J.; Xia, Y.; Gu, N.; Zhang, F., Magnetic Cell-Scaffold Interface Constructed by Superparamagnetic IONP Enhanced Osteogenesis of Adipose-Derived Stem Cells. Acs Applied Materials & Interfaces 2018, 10 (51), 44279-44289.
- Shu, Y.; Ma, M.; Pan, X.; Shafiq, M.; Yu, H.; Chen, H., Cobalt Protoporphyrin-Induced Nano-Self-Assembly for CT Imaging, Magnetic-Guidance, and Antioxidative Protection of Stem Cells in Pulmonary Fibrosis Treatment. Bioactive Materials 2023, 21, 129-141.
- Bakht, S. M.; Pardo, A.; Gomez-Florit, M.; Reis, R. L.; Domingues, R. M. A.; Gomes, M. E., Engineering Next-Generation Bioinks with Nanoparticles: Moving from Reinforcement Fillers to Multifunctional Nanoelements. Journal of Materials Chemistry B 2021, 9 (25), 5025-5038.
- “At present, the commonly used magnetic particles in magnetic biomaterials are iron powder, iron trioxide, iron tetroxide, and so on.” This sentence needs citation.
[Response]: Thanks for the comments. We added more relevant references (Li, et.al., hemphyschem 2018, 19 (16), 1965-1979; Yang, et.al., Cells 2022, 11 (20).) after “At present, the most commonly used magnetic particles in magnetic biomaterials are iron powder, iron trioxide, iron tetroxide, and so on.”.
[Action taken]: In page 5, line 13,we added reference no. 20, 91 after “At present, the most commonly used magnetic particles in magnetic biomaterials are iron powder, iron trioxide, iron tetroxide, and so on [20, 91].”
- Li, Y.; Ye, D.; Li, M.; Ma, M.; Gu, N., Adaptive Materials Based on Iron Oxide Nanoparticles for Bone Regeneration. Chemphyschem 2018, 19 (16), 1965-1979.
- Yang, J.; Wu, J.; Guo, Z.; Zhang, G.; Zhang, H., Iron Oxide Nanoparticles Combined with Static Magnetic Fields in Bone Remodeling. Cells 2022, 11 (20).
- For biomedical applications:
The toxicity and availability of the fabricated materials should be clear. A subsection related to toxicity should be inserted to clear this point.
[Response]: Thank you for your cogitative question. Indeed, the safety of SPIONs is of vital importance for biomedical applications in vivo. We added a subsection related to the toxicity of materials in the application section. The toxicity of SPIONs is very small. Although organic chemicals are used in the preparation process, the synthesized SPIONs has good hydrophilicity and biocompatibility. Moreover, some SPIONs prepared by the classic chemical co-precipitation method have been approved by the FDA for clinical use (doi.org/10.1007/s11095-016-1958-5).
[Action taken]: In Page 14, line 41, we added the following description “3.1. Toxicity
SPIONs are widely used in bone tissue engineering; therefore, the toxicity of these material has aroused great concern. In recent years, there have been many studies on the toxicity of SPIONs, but according to the research, their toxicity is relatively small[27, 128]. Although organic chemicals are used in the preparation process, the synthesized SPIONs showed good hydrophilicity and biocompatibility[47, 129]. For example, some SPIONs prepared by the classic chemical co-precipitation method have been approved by the FDA for clinical use[130]. It has been found that SPIONs usually accumulate at a high level in the kidneys and organs of the reticuloendothelial system, including the liver, spleen, and bone marrow[131]. A study showed that a dose of SPIONs exceeding 35 mg/kg will cause significant toxicity to the liver and kidneys, indicating that the toxicity is dose-dependent[132]. The metal material in SPIONs cannot be cleared by the body because they are non-biodegradable particles[27]. Similarly, the long-term complete elimination of SPIONs is also uncertain. It has been found that the clearance of SPIONs obviously depends on the dose, and higher doses were proven to take longer to completely clear[44, 133, 134].”.
- Samrot, A. V.; Sahithya, C. S.; Selvarani A, J.; Purayil, S. K.; Ponnaiah, P., A Review on Synthesis, Characterization and Potential Biological Applications of Superparamagnetic Iron Oxide Nanoparticles. Current Research in Green and Sustainable Chemistry 2021, 4, 100042.
- Dasari, A.; Xue, J.; Deb, S., Magnetic Nanoparticles in Bone Tissue Engineering. Nanomaterials 2022, 12 (5).
- Gu, N.; Zhang, Z.; Li, Y., Adaptive Iron-based Magnetic Nanomaterials of High Performance for Biomedical Applications. Nano Research 2022, 15 (1), 1-17.
- Li, L.; Jiang, L.-L.; Zeng, Y.; Liu, G., Toxicity of Superparamagnetic Iron Oxide Nanoparticles: Research Strategies and Implications for Nanomedicine. Chinese Physics B 2013, 22 (12).
- Lu, A.-H.; Salabas, E. L.; Schueth, F., Magnetic Nanoparticles: Synthesis, Protection, Functionalization, and Application. Angewandte Chemie-International Edition 2007, 46 (8), 1222-1244.
- Bobo, D.; Robinson, K. J.; Islam, J.; Thurecht, K. J.; Corrie, S. R., Nanoparticle-Based Medicines: A Review of FDA-Approved Materials and Clinical Trials to Date. Pharmaceutical Research 2016, 33 (10), 2373-2387.
- Liu, G.; Gao, J.; Ai, H.; Chen, X., Applications and Potential Toxicity of Magnetic Iron Oxide Nanoparticles. Small 2013, 9 (9-10), 1533-1545.
- Ma, P.; Luo, Q.; Chen, J.; Gan, Y.; Du, J.; Ding, S.; Xi, Z.; Yang, X., Intraperitoneal Injection of Magnetic Fe3O4-Nanoparticle Induces Hepatic and Renal Tissue Injury via Oxidative Stress in Mice. International Journal of Nanomedicine 2012, 7, 4809-4818.
- Storey, P.; Lim, R. P.; Chandarana, H.; Rosenkrantz, A. B.; Kim, D.; Stoffel, D. R.; Lee, V. S., MRI Assessment of Hepatic Iron Clearance Rates After USPIO Administration in Healthy Adults. Investigative Radiology 2012, 47 (12), 717-724.
- Jarockyte, G.; Daugelaite, E.; Stasys, M.; Statkute, U.; Poderys, V.; Tseng, T.-C.; Hsu, S.-H.; Karabanovas, V.; Rotomskis, R., Accumulation and Toxicity of Superparamagnetic Iron Oxide Nanoparticles in Cells and Experimental Animals. International Journal of Molecular Sciences 2016, 17 (8).
- Recent citations on magnetic nanomaterial for biomedical applications should include supporting the review
https://doi.org/10.3390/ma15217823
https://doi.org/10.3390/ijms23031023
https://doi.org/10.3390/magnetochemistry7100133
[Response]: Thanks for the comments. Those citations are very important for our research and give us many ideas. We refer and cite these relevant article (https://doi.org/10.3390/ma15217823; https://doi.org/10.3390/ijms23031023; https://doi.org/10.3390/magnetochemistry7100133) in the article.
[Action taken]: In page 1, line 39, we added reference no. 30 after “…superparamagnetic iron oxide nanoparticles (SPION) have achieved fruitful results in the fields of biosensors, non-invasive cell tracking, immune detection, drug delivery, and tumor diagnosis and treatment[23-30].” In page 1, line 45, we added reference no. 38 after “Thanks to the important properties of SPIONs, such as their good biocompatibility and superparamagnetic behavior, they have become an excellent choice for magnetic thermo-therapy and as a magnetic resonance imaging contrast agent[27, 35-40].” In page 7, line 15, we added reference no. 98 after “Electrospinning is a powerful, quite simple, and widely used process. Electrospinning involves a jet erupting from the tip of a spinneret to produce fibers with diameters as low as the submicron or nanometer scales [96-98].”
- Marino, M. A.; Fulaz, S.; Tasic, L., Magnetic Nanomaterials as Biocatalyst Carriers for Biomass Processing: Immobilization Strategies, Reusability, and Applications. Magnetochemistry 2021, 7 (10).
- Miyamoto, Y.; Koshidaka, Y.; Murase, K.; Kanno, S.; Noguchi, H.; Miyado, K.; Ikeya, T.; Suzuki, S.; Yagi, T.; Teramoto, N.; Hayashi, S., Functional Evaluation of 3D Liver Models Labeled with Polysaccharide Functionalized Magnetic Nanoparticles. Materials 2022, 15 (21).
- Darwish, M. S. A.; Mostafa, M. H.; Al-Harbi, L. M., Polymeric Nanocomposites for Environmental and Industrial Applications. International Journal of Molecular Sciences 2022, 23 (3).
- Summary and prospect:
There is a lack of critical assessments by the authors. The authors did not mention the research gap between the previously reported articles and the present situation. Authors should incorporate their views to mold the research in a new direction.
[Response]: Thanks for your suggestion of prospective significance. We have integrated the views of other authors and add a new direction. We expand a bit more the outlook section. The magnetically-assisted 3D bioprinting technology is mentioned, and the potential applications of magnetoceramic nanoparticles are commented. Magnetically assisted 3D bioprinting techniques can be exploited to design magnetically responsive, cell-laden scaffolds with impressive control over their resolution and shape fidelity, broadening the available design space of hybrid magnetic composites. Magnetoceramic compositions are also promising materials with high magnetic properties that could promote further development in bone tissue engineering.
[Action taken]: We added the following description “Magnetic scaffolds may produce uncontrolled aggregation in the biological environment or release metal ions that are potentially toxic to cells[50]. Therefore, designing highly monodisperse functionalized SPIONs can prevent uncontrolled aggregation from occurring. Moreover, magnetic scaffolds with high magnetic energy contain low amounts of magnetic material and require lower intensities of magnetic radiation for remote control, reducing the related toxicity[50]. However, there are still many challenges related to the development of magnetic scaffolds in the bone tissue engineering field. This usually requires the use of novel processing techniques. For example, magnetically assisted 3D bioprinting techniques can be exploited to design magnetically responsive, cell-laden scaffolds with impressive control over their resolution and shape fidelity, broadening the available design space of hybrid magnetic composites[119]. Magnetoceramic compositions are also promising materials with high magnetic properties that could promote further development in bone tissue engineering[146].”.
- Pardo, A.; Bakht, S. M.; Gomez-Florit, M.; Rial, R.; Monteiro, R. F.; Teixeira, S. P. B.; Taboada, P.; Reis, R. L.; Domingues, R. M. A.; Gomes, M. E., Magnetically-Assisted 3D Bioprinting of Anisotropic Tissue-Mimetic Constructs. Advanced Functional Materials 2022.
- Bakht, S. M.; Pardo, A.; Gomez-Florit, M.; Reis, R. L.; Domingues, R. M. A.; Gomes, M. E., Engineering Next-Generation Bioinks with Nanoparticles: Moving from Reinforcement Fillers to Multifunctional Nanoelements. Journal of Materials Chemistry B 2021, 9 (25), 5025-5038.
- Nam, H.-G.; Huh, T.-H.; Kim, M.; Kim, J.; Kwark, Y.-J., Magnetic Properties of Amorphous Silicon Carbonitride-based Magnetoceramics Synthesized Using Phenyl-Substituted Polysilazane as a Precursor. Journal of Alloys and Compounds 2022, 905.
- References
Should have one system style
[Response]: Thanks for the comments. We unified the reference style.
[Action taken]: We unified the reference style. For example, 1. Zhang, L.; Yang, G.; Johnson, B. N.; Jia, X., Three-Dimensional (3D) Printed Scaffold and Material Selection for Bone Repair. Acta Biomaterialia 2019, 84, 16-33.

Reviewer 2 Report
In the paper entitled “Biological Scaffolds Assembled with Magnetic Iron Oxide Nanoparticles for Bone Tissue Engineering: A Review” the authors reviewed the main methods for the fabrication of magnetically-responsive scaffolds and their application in the fields of bone and cartilage repair.
In general, I found that major revisions are required before considering the manuscript suitable for its publication in Material journal. The introduction should be completed to properly address the state of the art and the significance of the proposed revision. Moreover, although I am not a native English speaker, the used English style should be professionally reviewed. Following, I expose some comments and suggestions that could improve the paper and which I would like the authors to address before consider resubmission:
I suggest removing “magnetic” or, even better, “iron oxide” from the title. “Biological Scaffolds Assembled with Magnetic Nanoparticles for Bone Tissue Engineering: A Review” sound better in my opinion.
“In addition, biological scaffolds usually have good biocompatibility, and can have a variety of different properties and properties through design, assembly, surface modification, and other processes, so it has become the focus of medical research”. Confusing sentence, rephrase it.
“At present, the commonly used superparamagnetic iron oxide nanoparticles (SPION) is a kind of crystalline material composed of magnetic iron oxide.” I suggest to remove this sentence, it is obvious that superparamagnetic iron oxide NPs are compose by iron oxide.
“Thanks to its important properties such as non-toxic, good biocompatibility, and high aggregation, SPION has become an excellent choice for magnetic thermo-therapy and magnetic resonance imaging contrast agent”. This sentence should be rephrased. The potential aggregation of magnetic nanoparticles is a problem in view of their potential application in the mentioned fields such as magnetic hyperthermia or MRI (see doi: 10.1021/acsami.9b20496 and 10.1021/acs.chemmater.9b04848). It is the reason why magnetic nanoparticles with superparamagnetic behavior are typically required for biomedical purposes, because they do not retain any magnetization after remove the applied magnetic field thus preventing their incontrolled agglomeration
Thus this review is focused on bone tissue engineering; I miss a short description of bone tissues in the introduction section: architecture, mechanical properties, characteristics of cells, etc. How SPIONs help to replicate this characetristics in engineered scaffolds? You can take a look in the last section of the recent review article about magnetic nanocomposite hydrogels published by Manuela E. Gomes et al. in ACS Nano, where before discussing magnetic strategies for bone tissue engineering they briefly described bone tissues. In this way, you can ellaborate a bit that MNPs can be used to increase the stiffness and other mechanical properties of polymeric hydrogels typically used for bone TE, thus more closely recapitulate the mechanical properties of native bones. MNPs can also be arranged in the scaffolds replicatinng the anisotropic architectures of natural bones or for the magnetomechanical stimulations of engieered scaffolds during maturation process, which has been demostrated as an essential point to replicate the native dynamic cellular environments.
Please also mention here in the intro the most typical materials used for bone tissue engineering (e.g. polymeric hydrogels, elctrospun fibers) and also refer the most typical modification based on the incorporatio of ceramic nanoparticles (hydroxyapatite) to replicate composition of native bones
I suggest to incorporate a brief discussion about the sythesis of magnetic nanoparticles and the importance of design magnetic nanostructures with high magnetic response (large magnetization values). In this way, the fabricated scaffolds can be provided with magnetic response bay incorporating low amounnts of magnetic material and then remotely manipulated by applying low-intensity magnetic fields, thus minimizing the toxicity/safety risks associated with these factors. Besides the previously suggested paper, you can read and refer other article recently published by the 3B’s research group (doi: 10.1002/adfm.202208940). Here they discuss the importance of design highly-responsive magnetic nanoparticles for their subsequent icorporation in tissue egineered scaffolds, in this case for tendon regeneration. In previous works of their collaborators (e.g. https://link.springer.com/chapter/10.1007/978-3-319-89878-0_7) you can check and mention the main design strategies to provide magnetic nnanoparticles with this desired high magnetic response (control of size, morphology, composition, structure, etc.).
In third column of Table 1 I suggest to remove the successive “nanoparticles”. This word is already indicated in the top file, so you can entitle this column “NPs composition” or something like that and remove NPs from all the files.
“Magnetic scaffold is one of the magnetic materials, which is widely studied in recent years, which mainly covers the materials with magnetic and biological properties” Rephrase sentence, it is not well-constructed with these double “magnetic” and double “which”.
“The magnetic field in the material is used as a carrier to assemble and apply to biomaterials, such as disease diagnosis, drug treatment, MRI imaging, and other functions”. Remove this sentence; you already mentioned these other applications of magnetic materials in the itro. Here it makes no sense, focus on the topic of the review: MNPs for bone tissue engineering.
Reference [44] is placed between “Frank” and “et al”. Change it. Apply throughout the mauscript.
“Díaz[47] was also freeze-dried to prepare polycaprolactone/hydroxyapatite (PCL/HA) magnetic composite scaffolds with different composition”. Rephrase please
In general I found that the continuous referring to other Works inn the form of “YYY et al…” makes the manuscript a bit repetitive. Use other and alterative styles to start the description of other works would be better in my opinion.
“…which can re-alize the injection of electric fluid from the process solution or melt…” Rephrase this part of the sentence.
I think that Figures 3, 4, and 5 can be mixed in only one bigger panel devoted to these three fabrication techniques. Maybe it would be more appropiate in a review article like this. On the other hand, increase the resolution of all the figures, please.
Typo in 3D printing section: “3.5nm”
In printing section at least you shoul mention 3D bio-printing techniques, where cells are incorporated in the bioinks previous to their manufacturig to design cell-laden scaffolds. You can see several representative examples of this technique in https://doi.org/10.1039/D1TB00717C. In this review you can also found representative examples of the combination of 3D bioprinting with magnetic materials to fabricate cell-laden magnetic hydrogels. Applications in the field of bonne tissue egineering are also described in this article
Section 2.4: SF abbreviation is used without be defined previously.
“Thanks to the existence of SPION, the scaffold shows excellent thermotherapy performance under alter-nating magnetic field, which can improve the adhesion and colonization of osteoblasts” How local icrements of temperature caused by the presence of MNPs under the application of an externnal magnetic field (magnnetic hyperthemia) improve cells adhesion? Ellaborate a little bit, please.
“At the same time, because of its superparamagnetism, the composite scaffold can generate heat in the alternating magnetic field and increase the temperature of the surrounding environment, which is helpful to its application in bone tissue engineering.” The generation of heat by MNPs is not due to their superparamagnetic behavior. You can write that it is due to their intrinsic magnetic reponse or something similar, but not due to their superparamagnetism.
Avoid the use of expressions such as “magnetic SPIONs” durinng the manuscript. Magnetic is already part of SPIONs definition.
“Angiogenesis is very important in osteogenesis, because the supply and oxygen sup-ply cannot be within 200 μm of the blood vessel, otherwise, without a good blood supply, the cells will not be able to survive and the formation of new bone will be hindered natu-rally.” Extremely cofusing sentence, rephrase.
Section 3.2: cartilage is a highly-anisotropic tissue, so here you can explain in detail how magnetic anoparticles can be incorporated in the biomaterials and then remotely arrannged in a controlled manner through the application of external magnetic fields to replicate in the scaffolds the annisotropic architecture of native cartilages (see for instance this work where this described strategy is combined with 3D bioprinting: 10.1002/adhm.201800894).
197 ± 10.3% cannot be expressed in this form. Main value and standard deviation must have the same degree of significance (197 ± 10 or 197.0 ± 10.3).
Sice you reviewed the application of magnetic scaffolds in bone and also in CARTILAGE tissue engineering, the title of section 3 (and even the title of the article) should include the name of both tissues.
I suggest expanding a bit more the outlook section. Mention novel and still low-explored processing techniques such as magnetically-assisted 3D bioprinting which can be exploited to design magnetically-responsive cell-laden scaffolds with impressive cotrol over their resolution and shape fidelity. You can also comment here the potential use of structures such as magnetoceramic nanoparticles, which have not been widely explored and are completely included in the scope of the review. Also ellaborate a bit more the biological concerns associated with the use of magnetic nanoparticles (uncontrolled aggregation in biological environments, long-term cytotoxicity for the release of potential toxic metallic ions, etc.)
Author Response
Reviewer #2:
Comments:
In the paper entitled “Biological Scaffolds Assembled with Magnetic Iron Oxide Nanoparticles for Bone Tissue Engineering: A Review” the authors reviewed the main methods for the fabrication of magnetically-responsive scaffolds and their application in the fields of bone and cartilage repair.
In general, I found that major revisions are required before considering the manuscript suitable for its publication in Material journal. The introduction should be completed to properly address the state of the art and the significance of the proposed revision. Moreover, although I am not a native English speaker, the used English style should be professionally reviewed. Following, I expose some comments and suggestions that could improve the paper and which I would like the authors to address before consider resubmission:
[Response]: Thank you very much for your kind comments and your patience on our work. We have made significant changes to the article. We are sorry for our unprofessional English, and now the manuscript have been checked and edited by a native English-speaking expert from MDPI office.
- I suggest removing “magnetic” or, even better, “iron oxide” from the title. “Biological Scaffolds Assembled with Magnetic Nanoparticles for Bone Tissue Engineering: A Review” sound better in my opinion.
[Response]: Thanks for the comments. We changed the title to Biological Scaffolds Assembled with Magnetic Nanoparticles for Bone Tissue Engineering: A Review.
[Action taken]: On Page 1, we changed the title “Biological Scaffolds Assembled With Magnetic Iron Oxide Nanoparticles For Bone Tissue Engineering: A Review” to “Biological Scaffolds Assembled with Magnetic Nanoparticles for Bone Tissue Engineering: A Review”.
- “In addition, biological scaffolds usually have good biocompatibility, and can have a variety of different properties and properties through design, assembly, surface modification, and other processes, so it has become the focus of medical research”. Confusing sentence, rephrase it.
[Response]: Thanks for the comments. We rephrase this sentence.
[Action taken]: In Page 1, line 29, we changed the following descriptions “In addition, biological scaffolds usually have good biocompatibility, and can have a variety of different properties and properties through design, assembly, surface modification, and other processes, so it has become the focus of medical research” to “In addition, scaffolds usually have good biocompatibility and can have a variety of different properties depending on their design, assembly, and surface modification; they have therefore become a focus in medical research.”
- “At present, the commonly used superparamagnetic iron oxide nanoparticles (SPION) is a kind of crystalline material composed of magnetic iron oxide.” I suggest to remove this sentence, it is obvious that superparamagnetic iron oxide NPs are compose by iron oxide.
[Response]: Thanks for the comments. We remove this sentence
[Action taken]: In Page 1, line 38, we removed the following description “At present, the commonly used superparamagnetic iron oxide nanoparticles (SPION) is a kind of crystalline material composed of magnetic iron oxide.”.
- “Thanks to its important properties such as non-toxic, good biocompatibility, and high aggregation, SPION has become an excellent choice for magnetic thermo-therapy and magnetic resonance imaging contrast agent”. This sentence should be rephrased. The potential aggregation of magnetic nanoparticles is a problem in view of their potential application in the mentioned fields such as magnetic hyperthermia or MRI (see doi: 10.1021/acsami.9b20496and 10.1021/acs.chemmater.9b04848). It is the reason why magnetic nanoparticles with superparamagnetic behavior are typically required for biomedical purposes, because they do not retain any magnetization after remove the applied magnetic field thus preventing their incontrolled agglomeration
[Response]: Thanks for the comments. We rephrase this sentence. Moreover, we refer and cite two papers (doi: 10.1021/acsami.9b20496 and 10.1021/acs.chemmater.9b04848).
[Action taken]: In Page 1, line 45, we changed the following descriptions “Thanks to its important properties such as non-toxic, good biocompatibility, and high aggregation, SPION has become an excellent choice for magnetic thermo-therapy and magnetic resonance imaging contrast agent” to “Thanks to the important properties of SPIONs, such as their good biocompatibility and superparamagnetic behavior, they have become an excellent choice for magnetic thermo-therapy and as a magnetic resonance imaging contrast agent[27, 35-40].”.
- Pardo, A.; Yanez, S.; Pineiro, Y.; Iglesias-Rey, R.; Al-Modlej, A.; Barbosa, S.; Rivas, J.; Taboada, P., Cubic Anisotropic Co- and Zn-Substituted Ferrite Nanoparticles as Multimodal Magnetic Agents. Acs Applied Materials & Interfaces 2020, 12 (8), 9017-9031.
- Pardo, A.; Pelaz, B.; Gallo, J.; Banobre-Lopez, M.; Parak, W. J.; Barbosa, S.; del Pino, P.; Taboada, P., Synthesis, Characterization, and Evaluation of Superparamagnetic Doped Ferrites as Potential Therapeutic Nanotools. Chemistry of Materials 2020, 32 (6), 2220-2231.
- Thus this review is focused on bone tissue engineering; I miss a short description of bone tissues in the introduction section: architecture, mechanical properties, characteristics of cells, etc. How SPIONs help to replicate this characetristics in engineered scaffolds? You can take a look in the last section of the recent review article about magnetic nanocomposite hydrogels published by Manuela E. Gomes et al. in ACS Nano, where before discussing magnetic strategies for bone tissue engineering they briefly described bone tissues. In this way, you can ellaborate a bit that MNPs can be used to increase the stiffness and other mechanical properties of polymeric hydrogels typically used for bone TE, thus more closely recapitulate the mechanical properties of native bones. MNPs can also be arranged in the scaffolds replicatinng the anisotropic architectures of natural bones or for the magnetomechanical stimulations of engieered scaffolds during maturation process, which has been demostrated as an essential point to replicate the native dynamic cellular environments.
[Response]: Thanks for your suggestion, which is important value for us deep thinking and elucidating this point. We added a short description of bone tissues in the introduction section: architecture, mechanical properties, characteristics of cells, etc. Natural bone tissue has complex hierarchical structures with varying scales of length and width composed of trabecular bone, haversian canals, osteons, lamellae, fibrillar collagen, minerals, collagen, and so on. The mechanical properties of natural bones are different in different body parts. However, the longitudinal direction of compact bone is stronger than its transverse direction in bone tissue. Moreover, natural bone contains cells, extracellular matrices, and bound minerals. We refer and cite the recent review article about magnetic nanocomposite hydrogels published by Manuela E. Gomes et al. in ACS Nano. A scaffold containing SPIONs has structural features and functions close to those of natural bone, which can provide the scaffold with good biocompatibility, stiffness, and mechanical properties (Pardo, A, et.al., Acs Nano 2021, 15 (1), 175-209.). SPIONs can also be arranged in the scaffolds to replicate the anisotropic architectures of natural bones or for the magnetomechanical stimulation of engineered scaffolds during the maturation process, which has been demonstrated as essential in replicating native dynamic cellular environments (Pardo, A, et.al., Acs Nano 2021, 15 (1), 175-209.). SPIONs can make use of the superparamagnetism of magnetic compositions in cell microenvironments, enhance the osteogenesis and angiogenesis of scaffolds, and promote cell attachment, proliferation, and differentiation.
[Action taken]: In Page 2, line 7, we added the following description “Natural bone tissue has complex hierarchical structures with varying scales of length and width composed of trabecular bone, haversian canals, osteons, lamellae, fibrillar collagen, minerals, collagen, and so on[13, 45]. The mechanical properties of natural bones are different in different body parts. However, the longitudinal direction of compact bone is stronger than its transverse direction in bone tissue[13]. Moreover, natural bone contains cells, extracellular matrices, and bound minerals[13]. A scaffold containing SPIONs has structural features and functions close to those of natural bone, which can provide the scaffold with good biocompatibility, stiffness, and mechanical properties[18, 43, 44]. SPIONs can also be arranged in the scaffolds to replicate the anisotropic architectures of natural bones or for the magnetomechanical stimulation of engineered scaffolds during the maturation process, which has been demonstrated as essential in replicating native dynamic cellular environments[18]. SPIONs can make use of the superparamagnetism of magnetic compositions in cell microenvironments, enhance the osteogenesis and angiogenesis of scaffolds, and promote cell attachment, proliferation, and differentiation[18, 43, 44].”.
- Qu, H.; Fu, H.; Han, Z.; Sun, Y., Biomaterials for Bone Tissue Engineering Scaffolds: A Review. Rsc Advances 2019, 9 (45), 26252-26262.
- Pardo, A.; Gomez-Florit, M.; Barbosa, S.; Taboada, P.; Domingues, R. M. A.; Gomes, M. E., Magnetic Nanocomposite Hydrogels for Tissue Engineering: Design Concepts and Remote Actuation Strategies to Control Cell Fate. Acs Nano 2021, 15 (1), 175-209.
- Burdusel, A.-C.; Gherasim, O.; Andronescu, E.; Grumezescu, A. M.; Ficai, A., Inorganic Nanoparticles in Bone Healing Applications. Pharmaceutics 2022, 14 (4).
- Dasari, A.; Xue, J.; Deb, S., Magnetic Nanoparticles in Bone Tissue Engineering. Nanomaterials 2022, 12 (5).
- Barth, H. D.; Zimmermann, E. A.; Schaible, E.; Tang, S. Y.; Alliston, T.; Ritchie, R. O., Characterization of the Effects of x-Ray Irradiation on the Hierarchical Structure and Mechanical Properties of Human Cortical Bone. Biomaterials 2011, 32 (34), 8892-8904.
- Please also mention here in the intro the most typical materials used for bone tissue engineering (e.g. polymeric hydrogels, elctrospun fibers) and also refer the most typical modification based on the incorporatio of ceramic nanoparticles (hydroxyapatite) to replicate composition of native bones
[Response]: Thanks for your suggestion, which is important value for us deep thinking and elucidating this point. We mentioned here in the intro the most typical materials used for bone tissue engineering (e.g. polymeric hydrogels, elctrospun fibers) and also refered the most typical modification based on the incorporatio of ceramic nanoparticles (hydroxyapatite) to replicate composition of native bones.
[Action taken]: In Page 1, line 32, we added the following description “Materials with excellent properties, such as polymeric hydrogels and electrospun fibers, are being increasingly used in bone tissue engineering with the continuous development of various scaffolds. Moreover, it is common to replicate natural bones by modifying ceramic compositions (e.g., hydroxyapatite)[18].”.
- Pardo, A.; Gomez-Florit, M.; Barbosa, S.; Taboada, P.; Domingues, R. M. A.; Gomes, M. E., Magnetic Nanocomposite Hydrogels for Tissue Engineering: Design Concepts and Remote Actuation Strategies to Control Cell Fate. Acs Nano 2021, 15 (1), 175-209.
- I suggest to incorporate a brief discussion about the sythesis of magnetic nanoparticles and the importance of design magnetic nanostructures with high magnetic response (large magnetization values). In this way, the fabricated scaffolds can be provided with magnetic response bay incorporating low amounnts of magnetic material and then remotely manipulated by applying low-intensity magnetic fields, thus minimizing the toxicity/safety risks associated with these factors. Besides the previously suggested paper, you can read and refer other article recently published by the 3B’s research group (doi: 10.1002/adfm.202208940). Here they discuss the importance of design highly-responsive magnetic nanoparticles for their subsequent icorporation in tissue egineered scaffolds, in this case for tendon regeneration. In previous works of their collaborators (e.g. https://link.springer.com/chapter/10.1007/978-3-319-89878-0_7), you can check and mention the main design strategies to provide magnetic nnanoparticles with this desired high magnetic response (control of size, morphology, composition, structure, etc.).
[Response]: Thanks for your suggestion, which is important value for us deep thinking and elucidating this point. We incorporate a brief discussion about the sythesis of magnetic nanoparticles and the importance of design magnetic nanostructures with high magnetic response (large magnetization values). In this way, the fabricated scaffolds can be provided with magnetic response bay incorporating low amounnts of magnetic material and then remotely manipulated by applying low-intensity magnetic fields, thus minimizing the toxicity/safety risks associated with these factors (doi: 10.1002/adfm.202208940). Magnetic compositions with this desired high magnetic response can be obtained by controlling the size, morphology, composition, structure, and other factors of the structure (https://link.springer.com/chapter/10.1007/978-3-319-89878-0_7).
[Action taken]: In Page 2, line 29, we added the following description “In addition, it is important to design and synthesize magnetic nanostructures with a high magnetic response (large magnetization value). In this way, the fabricated scaffolds can be provided with a magnetic response by incorporating low amounts of magnetic material and then remotely manipulated by applying low-intensity magnetic fields, which could minimize the toxicity/safety risks associated with these factors and in turn increase the application potential of magnetic scaffolds[50]. Therefore, magnetic compositions with this desired high magnetic response can be obtained by controlling the size, morphology, composition, structure, and other factors of the structure [51].”.
- Pardo, A.; Bakht, S. M.; Gomez-Florit, M.; Rial, R.; Monteiro, R. F.; Teixeira, S. P. B.; Taboada, P.; Reis, R. L.; Domingues, R. M. A.; Gomes, M. E., Magnetically-Assisted 3D Bioprinting of Anisotropic Tissue-Mimetic Constructs. Advanced Functional Materials 2022.
- Polo, E.; del Pino, P.; Pardo, A.; Taboada, P.; Pelaz, B., Magnetic Nanoparticles for Cancer Therapy and Bioimaging. In Nanooncology: Engineering Nanomaterials for Cancer Therapy and Diagnosis, Goncalves, G.; Tobias, G., Eds. 2018; pp 239-279.
- In third column of Table 1 I suggest to remove the successive “nanoparticles”. This word is already indicated in the top file, so you can entitle this column “NPs composition” or something like that and remove NPs from all the files.
[Response]: Thanks for the comments. We remove the successive “nanoparticles” in third column of Table 1. We entitle this column “NPs composition”. We remove NPs from all the files.
[Action taken]: In Page 2, Table 1, We remove the successive “nanoparticles” in third column of Table 1. We entitle this column “NPs composition”. We remove NPs from all the files.
- “Magnetic scaffold is one of the magnetic materials, which is widely studied in recent years, which mainly covers the materials with magnetic and biological properties” Rephrase sentence, it is not well-constructed with these double “magnetic” and double “which”.
[Response]: Thanks for the comments. We rephrase this sentence.
[Action taken]: In Page 5, line 11, we changed the following descriptions “Magnetic scaffold is one of the magnetic materials, which is widely studied in recent years, which mainly covers the materials with magnetic and biological properties” to “Magnetic scaffolding is a biomaterials that has been widely studied in recent years, mainly those with magnetic and biological properties.”
- “The magnetic field in the material is used as a carrier to assemble and apply to biomaterials, such as disease diagnosis, drug treatment, MRI imaging, and other functions”. Remove this sentence; you already mentioned these other applications of magnetic materials in the itro. Here it makes no sense, focus on the topic of the review: MNPs for bone tissue engineering.
[Response]: Thanks for the comments. We removed this sentence.
[Action taken]: In Page 5, line 12, we removed the following description “The magnetic field in the material is used as a carrier to assemble and apply to biomaterials, such as disease diagnosis, drug treatment, MRI imaging, and other functions”.
- Reference [44] is placed between “Frank” and “et al”. Change it. Apply throughout the mauscript.
[Response]: Thanks for the comments. Throughout the article, we modified this format to Frank et al.[52]. We modified this format “YYY [Reference] et al.” to “YYY et al.[Reference]”
[Action taken]: In Page 6, line 5, we changed the following descriptions “Frank[44] et al. …” to “Frank et al.[52] …”. Other problematic formats are similar to this modification.
- “Díaz[47] was also freeze-dried to prepare polycaprolactone/hydroxyapatite (PCL/HA) magnetic composite scaffolds with different composition”. Rephrase please
[Response]: Thanks for the comments. We rephrase this sentence.
[Action taken]: In Page 6, line 42, we changed the following descriptions “Díaz[47] was also freeze-dried to prepare polycaprolactone/hydroxyapatite (PCL/HA) magnetic composite scaffolds with different composition” to “Díaz also used the freeze-drying method to prepare polycaprolactone/hydroxyapatite (PCL/HA) magnetic composite scaffolds with different compositions [57].”
- In general I found that the continuous referring to other Works inn the form of “YYY et al…” makes the manuscript a bit repetitive. Use other and alterative styles to start the description of other works would be better in my opinion.
[Response]: Thanks for the comments. We use other and alterative instead of “YYY et al…”, such as YYY and his collaborators, YYY and co-workers, YYY 's group, and so on.
[Action taken]: In the article, we use other and alterative instead of “YYY et al…”, such as YYY and his collaborators, YYY and co-workers, YYY 's group, and so on.
- “…which can re-alize the injection of electric fluid from the process solution or melt…” Rephrase this part of the sentence.
[Response]: Thanks for the comments. We rephrase this sentence.
[Action taken]: In Page 7, line 15, we changed the following descriptions “…which can realize the injection of electric fluid from the process solution or melt, and can easily produce various ultra-fine fibers or fiber structure polymers with diameters as low as submicron or nanometer.” to “…Electrospinning involves a jet erupting from the tip of a spinneret to produce fibers with diameters as low as the submicron or nanometer scales.”
- I think that Figures 3, 4, and 5 can be mixed in only one bigger panel devoted to these three fabrication techniques. Maybe it would be more appropiate in a review article like this. On the other hand, increase the resolution of all the figures, please.
[Response]: Thanks for the comments. We mixed Figures 3, 4 and 5 in only one bigger panel to illustrate these three fabrication techniques (electrospinning, three-dimensional (3D) printing, and chemical synthesis methods) and increased the resolution of all the figures to 500 pixels.
[Action taken]: In Page 13, we mixed Figures 3, 4 and 5 in only one bigger panel Figures 4 and increased the resolution of all the figures to 500 pixels.
.
Figure 4. (a) Characterization of a superparamagnetic nanofiber scaffold. I) Superparamagnetic nanofiber scaffold particles with a diameter of 5 mm. II) Schematic diagram of the scaffold particles implanted in the defect of L5 transverse process in rabbits. III) SEM image of the scaffold. Ⅳ) TEM image of the fibers in the scaffold. (Copyright 2013. Reproduced with permission from ref [63]. Springer Nature.) (b) Characterization of SPIONs–gelatin. I) Visible image of SPIONs–gelatin (FeCD-Gel) 3D printing scaffold doped with carbon quantum dots. II) Ultraviolet image of FeCD-Gel scaffold. III) SEM image and EDAX analysis of FeCD-Gel scaffold. Ⅳ) Image of FeCD-Gel scaffold implanted in subcutaneous rat model. (Copyright 2019. Reproduced with permission from ref [68]. ACS publication.) (c) Preparation of magnetic cellulose nanocrystal/dextran hydrogel and its application in artificial cartilage repair engineering. KGN is coupled on the surface of ultra-small superparamagnetic iron oxide. (Copyright 2019. Reproduced with permission from ref[83]. ACS publication.)
- Meng, J.; Xiao, B.; Zhang, Y.; Liu, J.; Xue, H.; Lei, J.; Kong, H.; Huang, Y.; Jin, Z.; Gu, N.; Xu, H., Super-Paramagnetic Responsive Nanofibrous Scaffolds Under Static Magnetic Field Enhance Osteogenesis for Bone Repair in Vivo. Scientific Reports 2013, 3.
- Das, B.; Girigoswami, A.; Dutta, A.; Pal, P.; Dutta, J.; Dadhich, P.; Srivas, P. K.; Dhara, S., Carbon Nanodots Doped Super-paramagnetic Iron Oxide Nanoparticles for Multimodal Bioimaging and Osteochondral Tissue Regeneration via External Magnetic Actuation. Acs Biomaterials Science & Engineering 2019, 5 (7), 3549-3560.
- Yang, W.; Zhu, P.; Huang, H.; Zheng, Y.; Liu, J.; Feng, L.; Guo, H.; Tang, S.; Guo, R., Functionalization of Novel Theranostic Hydrogels with Kartogenin-Grafted USPIO Nanoparticles To Enhance Cartilage Regeneration. Acs Applied Materials & Interfaces 2019, 11 (38), 34744-34754.
- Typo in 3D printing section: “3.5nm”
[Response]: Thanks for the comments. We corrected this typo.
[Action taken]: In Page 10, line 20, we changed the following descriptions “…3.5nm…” to “…3.5 nm…”
- In printing section at least you should mention 3D bio-printing techniques, where cells are incorporated in the bioinks previous to their manufacturing to design cell-laden scaffolds. You can see several representative examples of this technique in https://doi.org/10.1039/D1TB00717C. In this review you can also found representative examples of the combination of 3D bioprinting with magnetic materials to fabricate cell-laden magnetic hydrogels. Applications in the field of bonne tissue engineering are also described in this article
[Response]: Thanks for your suggestion, which is important value for us deep thinking and elucidating this point. In Three-Dimensional printing section, we mentioned 3D bio-printing techniques, where cells are incorporated in the bioinks previous to their manufacturing to design cell-laden scaffolds (doi.org/10.1039/D1TB00717C). We refer and cite the related examples of the combination of 3D bioprinting with magnetic materials to fabricate cell-laden magnetic hydrogels. As an example of such a system, a magnetic bio-ink based on alginate and methylcellulose with incorporated magnetite microparticles was produced and shown to be highly compatible with the encapsulation of human mesenchymal stem cell line (hMSC), thus promoting the development of 3D bioprinting technology (doi.org/10.1021/acsbiomaterials.0c01371).
[Action taken]: In Page 10, line 29, we added the following description “At present, 3D bioprinting technology is attracting great attention in the field of bone tissue engineering, and cells are being incorporated into bio-ink before manufacturing to produce scaffolds loaded with cells[119]. As an example of such a system, a magnetic bio-ink based on alginate and methylcellulose with incorporated magnetite microparticles was produced and shown to be highly compatible with the encapsulation of human mesenchymal stem cell line (hMSC), thus promoting the development of 3D bioprinting technology[70].”.
- Spangenberg, J.; Kilian, D.; Czichy, C.; Ahlfeld, T.; Lode, A.; Gunther, S.; Odenbach, S.; Gelinsky, M., Bioprinting of Magnetically Deformable Scaffolds. Acs Biomaterials Science & Engineering 2021, 7 (2), 648-662.
- Bakht, S. M.; Pardo, A.; Gomez-Florit, M.; Reis, R. L.; Domingues, R. M. A.; Gomes, M. E., Engineering Next-Generation Bioinks with Nanoparticles: Moving from Reinforcement Fillers to Multifunctional Nanoelements. Journal of Materials Chemistry B 2021, 9 (25), 5025-5038.
- Section 2.4: SF abbreviation is used without be defined previously.
[Response]: Thanks for the comments. We redefined SF abbreviation. SF is the abbreviation of silk fibroin.
[Action taken]: In Page 11, line 1, we changed the following descriptions “SF…” to “Silk fibroin (SF) …”
- “Thanks to the existence of SPION, the scaffold shows excellent thermotherapy performance under alter-nating magnetic field, which can improve the adhesion and colonization of osteoblasts” How local icrements of temperature caused by the presence of MNPs under the application of an external magnetic field (magnetic hyperthemia) improve cells adhesion? Elaborate a little bit, please.
[Response]: Thanks for the comments. Thanks to the presence of SPIONs, the magnetic scaffolds showed excellent hyperthermia properties under alternating magnetic field. However, cell adhesion improved is caused by the good biocompatibility of the scaffold. We rephrase this sentence.
[Action taken]: In Page 11, line 9, we changed the following descriptions “Thanks to the existence of SPION, the scaffold shows excellent thermotherapy performance under alternating magnetic field, which can improve the adhesion and colonization of osteoblasts” to “Thanks to the presence of SPIONs, the magnetic scaffolds showed excellent hyperthermia properties under alternating magnetic field and were able raise the temperature to 8℃ in about 100 s. Moreover, the scaffold had good biocompatibility and improved the adhesion and colonization of osteoblasts.”
- “At the same time, because of its superparamagnetism, the composite scaffold can generate heat in the alternating magnetic field and increase the temperature of the surrounding environment, which is helpful to its application in bone tissue engineering.” The generation of heat by MNPs is not due to their superparamagnetic behavior. You can write that it is due to their intrinsic magnetic response or something similar, but not due to their superparamagnetism.
[Response]: Thanks for the comments. We change “its superparamagnetism” to “its intrinsic magnetic response”.
[Action taken]: In Page 11, line 38, we changed the following descriptions “At the same time, because of its superparamagnetism, the composite scaffold can generate heat in the alternating magnetic field and increase the temperature of the surrounding environment, which is helpful to its application in bone tissue engineering.” to “At the same time, due to its intrinsic magnetic response, the composite scaffold can generate heat in the alternating magnetic field and increase the temperature of the surrounding environment, which is helpful in its application in bone tissue engineering.”
- Avoid the use of expressions such as “magnetic SPIONs” durinng the manuscript. Magnetic is already part of SPIONs definition.
[Response]: Thanks for the comments. We modified all expression “magnetic SPIONs” to “SPION” during the manuscript.
[Action taken]: In the article, we modified all expression “magnetic SPIONs” to “SPION”.
- “Angiogenesis is very important in osteogenesis, because the supply and oxygen sup-ply cannot be within 200 μm of the blood vessel, otherwise, without a good blood supply, the cells will not be able to survive and the formation of new bone will be hindered natu-rally.” Extremely cofusing sentence, rephrase.
[Response]: Thanks for the comments. We rephrase this sentence.
[Action taken]: In Page 15, line 46, we changed the following descriptions “Angiogenesis is very important in osteogenesis, because the supply and oxygen supply cannot be within 200 μm of the blood vessel, otherwise, without a good blood supply, the cells will not be able to survive and the formation of new bone will be hindered naturally.” to “Angiogenesis is very important in osteogenesis because oxygen supply usually is no further than 200 μm away from blood vessels; otherwise, without good blood supply, the cells will not survive, and the formation of new bone will be hindered.”
- Section 3.2: cartilage is a highly-anisotropic tissue, so here you can explain in detail how magnetic anoparticles can be incorporated in the biomaterials and then remotely arrannged in a controlled manner through the application of external magnetic fields to replicate in the scaffolds the annisotropic architecture of native cartilages (see for instance this work where this described strategy is combined with 3D bioprinting: 10.1002/adhm.201800894).
[Response]: Thanks for the comments. Cartilage is a highly-anisotropic tissue. Therefore, here we explain in detail how magnetic scaffolds can be incorporated in the biomaterials and then remotely arranged in a controlled manner through the application of external magnetic fields to replicate the anisotropic architecture of native cartilages in the scaffolds. We refer and cite this paper (10.1002/adhm.201800894). Magnetic collagen agarose hydrogels were used to produce magnetic scaffolds by 3D bioprinting technology (10.1002/adhm.201800894). The magnetic scaffold can control the alignment of collagen fibers. The presence of magnetic particles and magnetic field triggered the alignment of collagen fibers in the desired direction, which obtained a bioprinted scaffold with alternating layers of aligned and random fibers. Based on this anisotropic structure, the magnetic composite scaffold enhanced mechanical stiffness and expressed more collagen II compared with single-layer materials, improving the potential of bioprinted scaffolds in cartilage regeneration.
[Action taken]: In Page 17, line 15, we added the following description “Due to the highly anisotropic tissue of cartilage, magnetic scaffolds can be incorporated in the biomaterials and then remotely arranged in a controlled manner through the application of external magnetic fields to replicate the anisotropic architecture of native cartilages in the scaffolds [143, 144]. For example, magnetic collagen agarose hydrogels were used to produce magnetic scaffolds by 3D bioprinting technology[145]. The magnetic scaffold can control the alignment of collagen fibers. The presence of magnetic particles and magnetic field triggered the alignment of collagen fibers in the desired direction, which obtained a bioprinted scaffold with alternating layers of aligned and random fibers. Based on this anisotropic structure, the magnetic composite scaffold enhanced mechanical stiffness and expressed more collagen II compared with single-layer materials, improving the potential of bioprinted scaffolds in cartilage regeneration.”.
- Di Bella, C.; Fosang, A.; Donati, D. M.; Wallace, G. G.; Choong, P. F. M., 3D Bioprinting of Cartilage for Orthopedic Surgeons: Reading between the Lines. Frontiers in Surgery 2015, 2.
- Campos, D. F. D.; Drescher, W.; Rath, B.; Tingart, M.; Fischer, H., Supporting Biomaterials for Articular Cartilage Repair. Cartilage 2012, 3 (3), 205-221.
- Betsch, M.; Cristian, C.; Lin, Y.-Y.; Blaeser, A.; Schoeneberg, J.; Vogt, M.; Buhl, E. M.; Fischer, H.; Campos, D. F. D., Incorporating 4D into Bioprinting: Real-Time Magnetically Directed Collagen Fiber Alignment for Generating Complex Multilayered Tissues. Advanced Healthcare Materials 2018, 7 (21).
- 197 ± 10.3% cannot be expressed in this form. Main value and standard deviation must have the same degree of significance (197 ± 10 or 197.0 ± 10.3).
[Response]: Thanks for the comments. We corrected this mistake.
[Action taken]: In Page 16, line 38, we changed the following descriptions “… and 197 ±10.3% respectively on the 14th day.” to “… and 197.0 ±10.3, respectively, on the 14th day.”
- Sice you reviewed the application of magnetic scaffolds in bone and also in CARTILAGE tissue engineering, the title of section 3 (and even the title of the article) should include the name of both tissues.
[Response]: Thanks for the comments. We change the title of section 3 to Application of magnetic scaffold in bone repair and cartilage repair.
[Action taken]: On Page 14, we changed the title of section 3 “Application of magnetic scaffold in bone tissue engineering” to “Application of magnetic scaffold in bone repair and cartilage repair”.
- I suggest expanding a bit more the outlook section. Mention novel and still low-explored processing techniques such as magnetically-assisted 3D bioprinting which can be exploited to design magnetically-responsive cell-laden scaffolds with impressive cotrol over their resolution and shape fidelity. You can also comment here the potential use of structures such as magnetoceramic nanoparticles, which have not been widely explored and are completely included in the scope of the review. Also ellaborate a bit more the biological concerns associated with the use of magnetic nanoparticles (uncontrolled aggregation in biological environments, long-term cytotoxicity for the release of potential toxic metallic ions, etc.)
[Response]: Thanks for your suggestion of prospective significance. We expand a bit more the outlook section. The magnetically-assisted 3D bioprinting technology is mentioned, and the potential applications of magnetoceramic nanoparticles are commented. Magnetically assisted 3D bioprinting techniques can be exploited to design magnetically responsive, cell-laden scaffolds with impressive control over their resolution and shape fidelity, broadening the available design space of hybrid magnetic composites. Magnetoceramic compositions are also promising materials with high magnetic properties that could promote further development in bone tissue engineering. In addition, we also emphasize the biological concerns associated with the use of magnetic nanoparticles (uncontrolled aggregation in biological environments, long-term cytotoxicity for the release of potential toxic metallic ions, etc.) Magnetic scaffolds may produce uncontrolled aggregation in the biological environment or release metal ions that are potentially toxic to cells. Therefore, designing highly monodisperse functionalized SPIONs can prevent uncontrolled aggregation from occurring. Moreover, magnetic scaffolds with high magnetic energy contain low amounts of magnetic material and require lower intensities of magnetic radiation for remote control, reducing the related toxicity.
[Action taken]: In Page 18, line 1, we added the following description “Magnetic scaffolds may produce uncontrolled aggregation in the biological environment or release metal ions that are potentially toxic to cells[50]. Therefore, designing highly monodisperse functionalized SPIONs can prevent uncontrolled aggregation from occurring. Moreover, magnetic scaffolds with high magnetic energy contain low amounts of magnetic material and require lower intensities of magnetic radiation for remote control, reducing the related toxicity[50]. However, there are still many challenges related to the development of magnetic scaffolds in the bone tissue engineering field. This usually requires the use of novel processing techniques. For example, magnetically assisted 3D bioprinting techniques can be exploited to design magnetically responsive, cell-laden scaffolds with impressive control over their resolution and shape fidelity, broadening the available design space of hybrid magnetic composites[119]. Magnetoceramic compositions are also promising materials with high magnetic properties that could promote further development in bone tissue engineering[146].”.
- Pardo, A.; Bakht, S. M.; Gomez-Florit, M.; Rial, R.; Monteiro, R. F.; Teixeira, S. P. B.; Taboada, P.; Reis, R. L.; Domingues, R. M. A.; Gomes, M. E., Magnetically-Assisted 3D Bioprinting of Anisotropic Tissue-Mimetic Constructs. Advanced Functional Materials 2022.
- Bakht, S. M.; Pardo, A.; Gomez-Florit, M.; Reis, R. L.; Domingues, R. M. A.; Gomes, M. E., Engineering Next-Generation Bioinks with Nanoparticles: Moving from Reinforcement Fillers to Multifunctional Nanoelements. Journal of Materials Chemistry B 2021, 9 (25), 5025-5038.
- Nam, H.-G.; Huh, T.-H.; Kim, M.; Kim, J.; Kwark, Y.-J., Magnetic Properties of Amorphous Silicon Carbonitride-based Magnetoceramics Synthesized Using Phenyl-Substituted Polysilazane as a Precursor. Journal of Alloys and Compounds 2022, 905.

Reviewer 3 Report
The paper can be accepted after a minor revision. Following are the comments:
1. In the introduction section, the effects of various factors such as shape, size, concentration, etc of SIONs in bone tissue engineering should be included.
2. Electrospinning is one of the versatile methods for preparing polymeric nanofibers/composites. Could the authors elaborate on this section by providing a schematic diagram for the electrospinning technique? Also, please highlight the various parameters of electrospinning. The authors can refer to and cite the following paper: https://doi.org/10.3390/pharmaceutics11070305
3. Here the authors have focused on the applications of SIONs for bone tissue engineering. How about the toxicity of the SIONs? The toxicity may be caused by SIONs themselves or may arise from the chemicals used during preparation.
Author Response
Reviewer #3:
Comments:
The paper can be accepted after a minor revision. Following are the comments:
[Response]: Thank you very much for your kind comments and your patience on our work.
- In the introduction section, the effects of various factors such as shape, size, concentration, etc of SIONs in bone tissue engineering should be included.
[Response]: Thanks for your suggestion, which is important value for us deep thinking and elucidating this point. In bone tissue engineering, various factors of SPIONs, such as shape, size, concentration, will also affect it. Therefore, in the introduction section, we add the influence of various factors such as the size, shape and concentration of SPIONs on bone tissue engineering.
[Action taken]: In Page 2, line 21, we added the following description “However, the shape and size of SPIONs can affect their application in bone tissue engineering[46, 47]. Only SPIONs with uniform surfaces and sizes less than 20 nm are considered to exhibit superparamagnetic behavior, that is, becoming permanently magnetized by an external magnetic field[27, 48]. In general, small SPIONs (< 10 nm) show a rapid metabolism while larger ones (> 200 nm) show a slow metabolism[49]. Moreover, the crystallinity and magnetic properties of SPIONs strongly influence their bioeffects in vivo[47]. Therefore, mastering the properties of SPIONs such as shape, size, concentration, and crystallinity can improve their combination with scaffolds in bone tissue engineering.”.
- Samrot, A. V.; Sahithya, C. S.; Selvarani A, J.; Purayil, S. K.; Ponnaiah, P., A Review on Synthesis, Characterization and Potential Biological Applications of Superparamagnetic Iron Oxide Nanoparticles. Current Research in Green and Sustainable Chemistry 2021, 4, 100042.
- Xiong, F.; Tian, J.; Hu, K.; Zheng, X.; Sun, J.; Yan, C.; Yao, J.; Song, L.; Zhang, Y.; Gu, N., Superparamagnetic Anisotropic Nano-Assemblies with Longer Blood Circulation in Vivo: A Highly Efficient Drug Delivery Carrier for Leukemia Therapy. Nanoscale 2016, 8 (39), 17085-17089.
- Gu, N.; Zhang, Z.; Li, Y., Adaptive Iron-based Magnetic Nanomaterials of High Performance for Biomedical Applications. Nano Research 2022, 15 (1), 1-17.
- Maleki, H.; Simchi, A.; Imani, M.; Costa, B. F. O., Size-Controlled Synthesis of Superparamagnetic Iron Oxide Nanoparticles and Their Surface Coating by Gold for Biomedical Applications. Journal of Magnetism and Magnetic Materials 2012, 324 (23), 3997-4005.
- Choi, H. S.; Liu, W.; Misra, P.; Tanaka, E.; Zimmer, J. P.; Ipe, B. I.; Bawendi, M. G.; Frangioni, J. V., Renal Clearance of Quantum Dots. Nature Biotechnology 2007, 25 (10), 1165-1170.
- Electrospinning is one of the versatile methods for preparing polymeric nanofibers/composites. Could the authors elaborate on this section by providing a schematic diagram for the electrospinning technique? Also, please highlight the various parameters of electrospinning. The authors can refer to and cite the following paper: https://doi.org/10.3390/pharmaceutics11070305
[Response]: Thanks for your suggestion of prospective significance. The nice reference gave us many ideas. According to your guidance, we have added the schematic diagram for the electrospinning technique and highlight the various parameters in the full text. The fiber morphology could be controlled by factors including solution parameters, process parameters and ambient into parameters. In detail, those factors include the concentration/viscosity/conductivity of the solution, the applied electric field, the flow rate, the humidity, the temperature and so on. These parameters all affect the morphology of fiber and none of them that act independently. Several reports have summarized the influences of parameters during the electrospinning. By introducing SPIONs to polymer fluid, magnetic nanofiber could be obtained. We refer and cite this paper (https://doi.org/10.3390/pharmaceutics11070305).
[Action taken]: In Page 7, line 19, we added the following description “A basic electrospinning device consists of four main parts: a high-voltage source, a syringe pump propulsion system, a spinneret, and a collector (Figure 3)[96, 101]. The main principle of electrospinning technology is that the electrostatic force is caused by the electric field generated by the high-voltage device. When the electrostatic force overcomes the surface tension of the solution ejected from the syringe needle, the droplets formed by the solution at the outlet of the needle will form a Taylor cone, and the trickle of the solution will be ejected from the outlet of the needle. In this process, the solution is gradually stretched and refined, the solvent evaporates, and the dried polymer is arranged on the collector to form nanofibers[102].”. In Page 8, line 13, we added the following description “In the process of electrospinning, the applied voltage, the distance between the spinneret and the collector, the forward speed of the solution, the temperature and humidity, the relative molecular weight of the polymer, the polymer concentration, the surface tension of the polymer solution, and the conductivity of the polymer solution will affect fiber formation[102, 104-107]. The impacts of these elements are described as follows. In terms of the applied voltage, increasing the voltage can make the spinning process easier. When the applied voltage increases, the fiber diameter will gradually decrease[108]. In terms of the distance between the spinneret and the collector, the receiving distance increases and the fiber diameter decreases; as the receiving distance decreases, the solvent cannot be completely volatilized in time, resulting in an uneven distribution of the spun fiber surface[108]. Keeping a moderate forward solution speed is beneficial to the formation of nanofibers. In a moderate range, with the increase in the forward speed of the solution, the diameter of the obtained nanofibers also increases[109]. Increases in temperature lead to increases in fiber diameter, and increases in humidity lead to decreases in the solvent volatilization rate[110, 111]. When the relative molecular weight of the polymer is low, nano-scale fiber materials cannot be formed[112]. When the polymer concentration of the solution is too high, the viscosity of the solution will increase, which will block the needle of the syringe and prevent normal spinning from occurring [108, 109]. In the process of electrospinning, if the surface tension of the charged droplets is too high and the force of the electrostatic field is less than the surface tension, the fibers will deform into droplets [113]. Increasing the conductivity of the polymer solution will accelerate the stretching of the solution, forming finer nanofibers[114].”.
Figure 3. Schematic diagram showing the electrospinning device.
- Ramakrishna, S.; Fujihara, K.; Teo, W.-E.; Yong, T.; Ma, Z.; Ramaseshan, R., Electrospun Nanofibers: Solving Global Issues. Materials Today 2006, 9 (3), 40-50.
- Wang, X.; Ding, B.; Sun, G.; Wang, M.; Yu, J., Electro-Spinning/Netting: A Strategy for the Fabrication of Three-Dimensional Polymer Nano-Fiber/Nets. Progress in Materials Science 2013, 58 (8), 1173-1243.
- Pant, B.; Park, M.; Park, S.-J., Drug Delivery Applications of Core-Sheath Nanofibers Prepared by Coaxial Electrospinning: A Review. Pharmaceutics 2019, 11 (7).
- Pant, B.; Park, M.; Park, S.-J.; Kim, H. Y., High Strength Electrospun Nanofiber Mats via CNT Reinforcement: A Review. Composites Research 2016, 29 (4), 186-193.
- Doshi, J.; Reneker, D. H., Electrospinning Process And Applications of Electrospun Fibers. Journal of Electrostatics 1995, 35 (2-3), 151-160.
- Pant, B.; Park, M.; Ojha, G. P.; Kim, D.-U.; Kim, H.-Y.; Park, S.-J., Electrospun Salicylic Acid/Polyurethane Composite Nanofibers for Biomedical Applications. International Journal of Polymeric Materials and Polymeric Biomaterials 2018, 67 (12), 739-744.
- Pant, B.; Ojha, G. P.; Kim, H.-Y.; Park, M.; Park, S.-J., Fly-Ash-Incorporated Electrospun Zinc Oxide Nanofibers: Potential Material for Environmental Remediation. Environmental Pollution 2019, 245, 163-172.
- Tijing, L. D.; Yao, M.; Ren, J.; Park, C.-H.; Kim, C. S.; Shon, H. K., Nanofibers for Water and Wastewater Treatment: Recent Advances and Developments. In Water and Wastewater Treatment Technologies, Bui, X. T.; Chiemchaisri, C.; Fujioka, T.; Varjani, S., Eds. 2019; pp 431-468.
- Zong, X. H.; Kim, K.; Fang, D. F.; Ran, S. F.; Hsiao, B. S.; Chu, B., Structure and Process Relationship of Electrospun Bioabsorbable Nanofiber Membranes. Polymer 2002, 43 (16), 4403-4412.
- Demir, M. M.; Yilgor, I.; Yilgor, E.; Erman, B., Electrospinning of Polyurethane Fibers. Polymer 2002, 43 (11), 3303-3309.
- Casper, C. L.; Stephens, J. S.; Tassi, N. G.; Chase, D. B.; Rabolt, J. F., Controlling Surface Morphology of Electrospun Polystyrene Fibers: Effect of Humidity and Molecular Weight in the Electrospinning Process. Macromolecules 2004, 37 (2), 573-578.
- Kailasa, S.; Reddy, M. S. B.; Maurya, M. R.; Rani, B. G.; Rao, K. V.; Sadasivuni, K. K., Electrospun Nanofibers: Materials, Synthesis Parameters, and Their Role in Sensing Applications. Macromolecular Materials and Engineering 2021, 306 (11).
- Haider, A.; Haider, S.; Kang, I.-K., A Comprehensive Review Summarizing the Effect of Electrospinning Parameters and Potential Applications of Nanofibers in Biomedical and Biotechnology. Arabian Journal of Chemistry 2018, 11 (8), 1165-1188.
- Yang, Q. B.; Li, Z. Y.; Hong, Y. L.; Zhao, Y. Y.; Qiu, S. L.; Wang, C.; Wei, Y., Influence of Solvents on the Formation of Ultrathin Uniform Poly(vinyl pyrrolidone) Nanofibers with Electrospinning. Journal of Polymer Science Part B-Polymer Physics 2004, 42 (20), 3721-3726.
- Here the authors have focused on the applications of SIONs for bone tissue engineering. How about the toxicity of the SIONs? The toxicity may be caused by SIONs themselves or may arise from the chemicals used during preparation.
[Response]: Thank you for your cogitative question. Indeed, the safety of SPIONs is of vital importance for biomedical applications in vivo. The toxicity of SPIONs is very small. Although organic chemicals are used in the preparation process, the synthesized SPIONs has good hydrophilicity and biocompatibility. Moreover, some SPIONs prepared by the classic chemical co-precipitation method have been approved by the FDA for clinical use (doi.org/10.1007/s11095-016-1958-5).
[Action taken]: In Page 14, line 41, we added the following description “3.1. Toxicity
SPIONs are widely used in bone tissue engineering; therefore, the toxicity of these material has aroused great concern. In recent years, there have been many studies on the toxicity of SPIONs, but according to the research, their toxicity is relatively small[27, 128]. Although organic chemicals are used in the preparation process, the synthesized SPIONs showed good hydrophilicity and biocompatibility[47, 129]. Moreover, the SIPONs prepared from chemical co-precipitation method showed better hydrophilicity and biocompatibility, which were prepared in aqueous solution[47]. For example, some SPIONs prepared by the classic chemical co-precipitation method have been approved by the FDA for clinical use[130]. It has been found that SPIONs usually accumulate at a high level in the kidneys and organs of the reticuloendothelial system, including the liver, spleen, and bone marrow[131]. A study showed that a dose of SPIONs exceeding 35 mg/kg will cause significant toxicity to the liver and kidneys, indicating that the toxicity is dose-dependent[132]. The metal material in SPIONs cannot be cleared by the body because they are non-biodegradable particles[27]. Similarly, the long-term complete elimination of SPIONs is also uncertain. It has been found that the clearance of SPIONs obviously depends on the dose, and higher doses were proven to take longer to completely clear[44, 133, 134].”.
- Samrot, A. V.; Sahithya, C. S.; Selvarani A, J.; Purayil, S. K.; Ponnaiah, P., A Review on Synthesis, Characterization and Potential Biological Applications of Superparamagnetic Iron Oxide Nanoparticles. Current Research in Green and Sustainable Chemistry 2021, 4, 100042.
- Dasari, A.; Xue, J.; Deb, S., Magnetic Nanoparticles in Bone Tissue Engineering. Nanomaterials 2022, 12 (5).
- Gu, N.; Zhang, Z.; Li, Y., Adaptive Iron-based Magnetic Nanomaterials of High Performance for Biomedical Applications. Nano Research 2022, 15 (1), 1-17.
- Li, L.; Jiang, L.-L.; Zeng, Y.; Liu, G., Toxicity of Superparamagnetic Iron Oxide Nanoparticles: Research Strategies and Implications for Nanomedicine. Chinese Physics B 2013, 22 (12).
- Lu, A.-H.; Salabas, E. L.; Schueth, F., Magnetic Nanoparticles: Synthesis, Protection, Functionalization, and Application. Angewandte Chemie-International Edition 2007, 46 (8), 1222-1244.
- Bobo, D.; Robinson, K. J.; Islam, J.; Thurecht, K. J.; Corrie, S. R., Nanoparticle-Based Medicines: A Review of FDA-Approved Materials and Clinical Trials to Date. Pharmaceutical Research 2016, 33 (10), 2373-2387.
- Liu, G.; Gao, J.; Ai, H.; Chen, X., Applications and Potential Toxicity of Magnetic Iron Oxide Nanoparticles. Small 2013, 9 (9-10), 1533-1545.
- Ma, P.; Luo, Q.; Chen, J.; Gan, Y.; Du, J.; Ding, S.; Xi, Z.; Yang, X., Intraperitoneal Injection of Magnetic Fe3O4-Nanoparticle Induces Hepatic and Renal Tissue Injury via Oxidative Stress in Mice. International Journal of Nanomedicine 2012, 7, 4809-4818.
- Storey, P.; Lim, R. P.; Chandarana, H.; Rosenkrantz, A. B.; Kim, D.; Stoffel, D. R.; Lee, V. S., MRI Assessment of Hepatic Iron Clearance Rates After USPIO Administration in Healthy Adults. Investigative Radiology 2012, 47 (12), 717-724.
- Jarockyte, G.; Daugelaite, E.; Stasys, M.; Statkute, U.; Poderys, V.; Tseng, T.-C.; Hsu, S.-H.; Karabanovas, V.; Rotomskis, R., Accumulation and Toxicity of Superparamagnetic Iron Oxide Nanoparticles in Cells and Experimental Animals. International Journal of Molecular Sciences 2016, 17 (8).

Round 2
Reviewer 2 Report
My comments and suggestions were well addressed by the authors, therefore I recommend the publication of the article